# Accelerating Model-Free Optimization
# via Averaging of Cost Samples

**Guido Carnevale**
Department of Electrical, Electronic and Information Engineering
Alma Mater Studiorum - Universita' di Bologna
Bologna, Italy
`guido.carnevale@unibo.it`

**Giuseppe Notarstefano**
Department of Electrical, Electronic and Information Engineering
Alma Mater Studiorum - Universita' di Bologna
Bologna, Italy
`giuseppe.notarstefano@unibo.it`

## Abstract

Model-free optimization methods typically rely on cost samples gathered by perturbing the current solution estimate along a finite and fixed set of directions. However, at each iteration, only the current cost samples are used, while potentially informative, previously collected samples are discarded. In this work, we challenge this conventional approach by introducing a simple yet effective memory mechanism that maintains an auxiliary vector of iteratively updated cost samples. By leveraging this stored information, our method estimates descent directions through an averaging of all perturbing directions weighted by the auxiliary vector components. This results in faster convergence without increasing the number of function queries. By interpreting the resulting algorithm as a time-varying dynamical system, we are able to establish its convergence properties in the strongly convex case. In particular, by using tools from system theory based on timescale separation, we are able to guarantee a linear convergence rate toward an arbitrarily small neighborhood of the optimal solution. Numerical simulations on regression problems demonstrate that the proposed approach significantly outperforms existing model-free optimization methods.

## 1 Introduction

At the core of many machine learning and deep learning tasks lie complex optimization problems that often cannot be solved analytically. In some cases, the objective function is not even known in closed form, allowing function evaluations but not gradient computations Liu et al. [2020]. The class of optimization methods that operate without analytical knowledge of the objective function (and thus without using its gradient) is referred to as model-free (or black-box) optimization. This branch of schemes Conn et al. [2009], Snoek et al. [2012], Larson et al. [2019] is receiving increasing attention in learning-oriented and related domains, such as signal processing Turner and Rasmussen [2012], reinforcement learning Malik et al. [2020], Qian and Yu [2021], IoT management Chen and Giannakis [2018], and system theory Galarza-Jimenez et al. [2022], He et al. [2023]. The most popular model-free optimization methods are the so-called zeroth-order methods Liu et al. [2020], in which the function evaluations are typically used to approximate its gradient. A distinction within this field involves the way in which the cost function is sampled, which can be done by using points that

39th Conference on Neural Information Processing Systems (NeurIPS 2025).

are selected in a random or deterministic fashion Kiefer and Wolfowitz [1952], Nesterov and Spokoiny [2017]. Another distinction is between 1-*point* Flaxman et al. [2005], Saha and Tewari [2011], Dekel et al. [2015], Gasnikov et al. [2017], Roy et al. [2022] and *multi-point* zeroth-order methods Agarwal et al. [2010], Duchi et al. [2015], Lian et al. [2016], Shamir [2017], Balasubramanian and Ghadimi [2022], based on the number of points selected at each iteration for cost evaluation. Indeed, in this field, cost function evaluations are typically expensive and, therefore, there is interest in reducing their number without compromising the convergence properties of the algorithm. Further, multi-point methods, besides being expensive, are often not practical, as they require simultaneous evaluations of the cost function at different points that are not possible in many applications, such as when the environment is non-stationary Hazan and Levy [2014], Bubeck et al. [2015], Yang and Mohri [2016], Zhao et al. [2021]. On the other hand, while 1-point zeroth-order methods are easily implementable and significantly reduce the number of evaluations, they typically suffer from slow convergence Liu et al. [2020]. To improve their performance, Chen et al. [2022] show the benefits of high-pass and low-pass filters. Xiao et al. [2023] introduce a suitable checking mechanism to possibly skip the cost function evaluation at each iteration. The works Zhang et al. [2022, 2024] use the residual between two feedback points at consecutive iterations to improve the performance of a single-point method.

Our main contribution is a novel algorithmic paradigm for model-free optimization. We assume that the unknown loss function can be evaluated by perturbing the current solution estimate along a finite, fixed set of directions. In this setting, we associate an auxiliary variable to each direction, which stores the corresponding cost sample as soon as it becomes available. Rather than updating the solution estimate solely based on the cost samples gathered in the current iteration, we estimate the descent direction by averaging all perturbation directions weighted by the auxiliary variables. This enables faster convergence without increasing the number of cost function queries. In the strongly convex case, by assuming a given gradient estimation technique and regular use of each perturbation direction, we formally show that the proposed method linearly converges to a neighborhood of the optimal solution whose radius can be made arbitrarily small by tuning the hyperparameters. Our line of proof interprets the resulting method as a time-varying dynamical system and leverages system-theoretic tools based on timescale separation between the memory mechanism and the solution update.

The paper unfolds as follows. In Section 2, we introduce the problem setup. In Section 3, we present the proposed algorithm and state its convergence guarantees. In Section 4, we analyze the proposed algorithm. Finally, in Section 5, we provide some numerical simulations on regression scenarios.

**Notation** The identity matrix of order $n$ is $I_n$. The all-zero vector in $\mathbb{R}^n$ is denoted as $0_n$. The vertical concatenation of column vectors $v_1, \ldots, v_N$ is $\mathrm{COL}(v_1, \ldots, v_N)$. The diagonal matrix with diagonal entries $a_1, \ldots, a_N \in \mathbb{R}$ is denoted as $\mathrm{diag}\{a_1, \ldots, a_N\} \in \mathbb{R}^N$.

## 2 Problem Setup

We consider unconstrained optimization problems of the form

$$\min_{\theta \in \mathbb{R}^n} \ell(\theta), \tag{1}$$

where $\ell : \mathbb{R}^n \to \mathbb{R}$ denotes the cost function. We focus on the following class of functions.

**Assumption 1.** *The cost function $\ell$ is $\mu$-strongly convex for some $\mu > 0$. Moreover, $\ell$ is differentiable and its gradient is $L$-Lipschitz continuous for some $L > 0$.* □

By Assumption 1, problem (1) admits a unique solution, denoted by $\theta_\star \in \mathbb{R}^n$. Our goal is to devise iterative methods to address problem (1) in a model-free scenario. In detail, we assume that the function $\ell$ is unknown but can be evaluated at each iteration (e.g., around the current solution estimate) to approximate its gradient according to a given, generic estimation technique introduced below.

### 2.1 Gradient Estimation Technique

Existing derivative-free methods in the literature properly sample the cost function $\ell$ to approximate the unavailable gradient $\nabla \ell$. A large part of these methods perform this sampling phase according to a *finite and fixed* set of additive perturbing (or dither) directions $\epsilon d^1, \ldots, \epsilon d^D \in \mathbb{R}^n$, where $\epsilon > 0$ is a parameter tuning the perturbation amplitude, while each $d^j \in \mathbb{R}^n$ characterizes the perturbation direction. More in detail, given a generic point $\theta \in \mathbb{R}^n$, the corresponding gradient $\nabla \ell(\theta)$ is estimated

by suitably manipulating $D \in \mathbb{N}$ cost samples $\ell(\theta + \epsilon d^1), \ldots, \ell(\theta + \epsilon d^D)$ gathered by perturbing $\theta$ along these directions. We formalize as follows the availability of a generic gradient estimation technique based on this sampling approach and we characterize its "ideal" estimation capabilities.

**Assumption 2.** *There exist $D \in \mathbb{N}$ directions $d^1, \ldots, d^D \in \mathbb{R}^n$ and a function $g^\epsilon : \mathbb{R} \times \mathbb{R}^n \to \mathbb{R}^n$ parametrized in $\epsilon > 0$ such that, for all compact sets $\mathcal{S} \subset \mathbb{R}^n$, there exists $\tilde{L} > 0$ such that*

$$\left\| \sum_{j=1}^{D} g^\epsilon(\ell(\theta + \epsilon d^j), d^j) - \nabla \ell(\theta) \right\| \leq \epsilon \tilde{L}, \tag{2}$$

*for all $\theta \in \mathcal{S}$ and $\epsilon > 0$.* $\qquad\square$

Assumption 2 characterizes the estimation capabilities of $\sum_{j=1}^{D} g^\epsilon(\ell(\theta + \epsilon d^j), d^j)$ in the ideal setting where $D$ queries of $\ell$ can be performed simultaneously. In detail, in such an ideal case, Assumption 2 ensures that the estimation technique yields an estimate of $\nabla \ell(\theta)$ (see (2)) whose accuracy can be made arbitrarily high by tuning the amplitude parameter $\epsilon$, which remains a free design choice in the algorithm. As one may expect, the perturbation directions $d^j$, their number $D$, the constant $\tilde{L}$, and the explicit structure of $g^\epsilon$ depend on the specific gradient estimation method that is employed.

### 2.1.1 Examples of Gradient Estimation Techniques

We provide some explicit examples about estimation techniques widely used in the literature that satisfy Assumption 2. In the schemes based on $2n$-point gradient estimators, the perturbation directions $d^1, \ldots, d^{2n}$ are given by the canonical basis $\pm\text{COL}(1, 0, \ldots, 0), \ldots, \pm\text{COL}(0, \ldots, 0, 1)$ in $\mathbb{R}^n$ and the corresponding estimation technique $g^\epsilon$ explicitly reads as

$$g^\epsilon(z, d) = zd/(2\epsilon), \tag{3}$$

for some $z \in \mathbb{R}$ (ideally, $z = \ell(\theta + \epsilon d)$, see Assumption 2). If the function $\ell$ has a Lipschitz continuous gradient, it can be shown that this technique satisfies Assumption 2 (see, e.g., Kiefer and Wolfowitz [1952] or Tang et al. [2020]). Extremum-seeking methods (see, e.g., the recent review Scheinker [2024] and the popular works Wittenmark and Urquhart [1995], Teel and Popovic [2001], Choi et al. [2002], Ariyur and Krstic [2003], Krstić and Wang [2000], Tan et al. [2006]) use sinusoidal perturbation directions. For instance, in the scalar case $n = 1$, we have $d^j = \sin(\frac{2\pi}{D} j + \phi)$ for all $j \in \{1, \ldots, D\}$ where $\phi \in \mathbb{R}$ is a phase shift. Thus, in these methods, the number of directions $D \in \mathbb{N}$ corresponds to the period of the sinusoidal signal, while the estimation function $g^\epsilon$ reads as

$$g^\epsilon(z, d) = 2zd/(\epsilon D), \tag{4}$$

for some $z \in \mathbb{R}$. If $\ell$ is $\mathcal{C}^3$ and with a proper choice of the sinusoidal functions' frequencies, it can be shown that this technique satisfies Assumption 2 (see, e.g., [Mimmo et al., 2024, Lemma 1]).

## 3 Averaged Model-Free Meta-Algorithm

In this section, we develop a novel method to address problem (1) in a model-free fashion. We propose it as a *meta-algorithm* in the sense that we show the updates of the solution estimate based on a new memory-mechanism paradigm and a generic gradient estimation oracle that can be made explicit in a specific algorithm (e.g., by using one of the schemes described in Section 2.1.1).

### 3.1 Meta-Algorithm Design

We remark that, in our model-free setting, it is not possible to simultaneously use all the perturbation directions $d^1, \ldots, d^D$ at each iteration. The idea behind the most popular 1-point zeroth-order methods (see Chen et al. [2022]) is instead to use only a single sample per iteration, with the rationale that, *on average*, the resulting descent direction approximately corresponds to the one characterized by (2) in Assumption 2. Mathematically, given the iteration index $t \in \mathbb{N}$, this would result in updating the current estimate $x_t \in \mathbb{R}^n$ about the solution to problem (1) according to the *time-varying* law

$$x_{t+1} = x_t - \gamma g^\epsilon(\ell(x_t + \epsilon d_t), d_t), \tag{5}$$

where $d_t \in \mathbb{R}^n$ is the direction used at iteration $t$ and $\gamma > 0$ is the step size. Instead, we pursue a different paradigm based on a memory mechanism that stores the queries $\ell(x_t + \epsilon d^1), \ldots, \ell(x_t + \epsilon d^D)$ whenever available. Namely, we use $D$ auxiliary variables $z_t^1, \ldots, z_t^D \in \mathbb{R}$ and update them as

$$z_{t+1}^j = \begin{cases} \ell(x_t + \epsilon d^j) & \text{if } \ell(x_t + \epsilon d^j) \text{ is gathered} \\ z_t^j & \text{otherwise,} \end{cases} \tag{6}$$

for all $j \in \{1, \ldots, D\}$. By introducing a selector signal $s_t^j \in \{0, 1\}$ for all $j \in \{1, \ldots, D\}$ to model whether $\ell(x_t + \epsilon d^j)$ is gathered or not at iteration $t$, we can equivalently rewrite (6) as

$$z_{t+1}^j = z_t^j + s_t^j(\ell(x_t + \epsilon d^j) - z_t^j).$$

Then, all these variables $z_{t+1}^j$ are used to approximate $\sum_{j=1}^D g^\epsilon(\ell(x_t + \epsilon d^j), d^j)$ (which, in turn, approximates the exact gradient $\nabla \ell(x_t)$, see Assumption 2) and update the solution estimate via

$$x_{t+1} = x_t - \gamma \sum_{j=1}^D g^\epsilon(z_{t+1}^j, d^j). \tag{7}$$

The overall model-free method arising from (6) and (7) is reported in Algorithm 1. We highlight that

---

**Algorithm 1** Averaged Model-Free Meta-Algorithm

---

  **Initialization**: $x_0 \in \mathbb{R}^n$, $z_0^1, \ldots, z_0^D \in \mathbb{R}$
  **for** $t = 0, 1, \ldots$ **do**
    **for** $j = 1, 2, \ldots, D$ **do**
      **if** $\ell(x_t + \epsilon d^j)$ is gathered, i.e., $s_t^j = 1$ **then**
        $z_{t+1}^j = \ell(x_t + \epsilon d^j)$
      **else**
        $z_{t+1}^j = z_t^j$
      **end if**
    **end for**
    $x_{t+1} = x_t - \gamma \sum_{j=1}^D g^\epsilon(z_{t+1}^j, d^j)$
  **end for**

---

Algorithm 1 does not necessarily increase the number of cost queries compared to 1-point approaches such as (5). Algorithm 1 uses the proxies $g^\epsilon(z_{t+1}^j, d^j)$ to mimic multiple queries at the price of an increased memory burden due to storing the $D$-dimensional vector $z_t := \text{COL}(z_t^1, \ldots, z_t^D) \in \mathbb{R}^D$.

### 3.2 Meta-Algorithm Convergence Properties

In this section, we state the convergence properties of Algorithm 1. To this end, we first introduce the following condition on the sampling sequences $\{s_t^1\}_{t \in \mathbb{N}}, \ldots, \{s_t^D\}_{t \in \mathbb{N}}$.

**Assumption 3.** *There exists $T_{max} \in \mathbb{N}$ such that, for all $t \in \mathbb{N}$ and $j \in \{1, \ldots, D\}$, there exists $\tau \in [t, t + T_{max} - 1]$ such that $s_\tau^j = 1$.* $\square$

Assumption 3 imposes an *essentially cyclic* selection of the directions $\{d^j\}_{t \in \mathbb{N}}$. Specifically, starting from all $t \in \mathbb{N}$, it ensures the existence of an upper bound $T_{\max}$ (independent of $t$) on the number of iterations required to select all directions at least once. We emphasize that Assumption 3 is very general and includes, as special cases, scenarios in which a fixed number of samples is collected at each iteration, ranging from a single sample to all $D$ samples without loss of generality. Moreover, our setting accommodates more challenging situations involving an irregular number of cost samples across iterations, including extreme cases where some iterations may involve no cost samples at all.

Now, we are ready to establish a linear convergence rate for Algorithm 1 towards a neighborhood of the optimal solution $\theta_\star$ with an arbitrarily small radius $\rho > 0$.

**Theorem 1.** *Let Assumptions 1, 2, and 3 hold. Then, for all $(x_0, z_0) \in \mathbb{R}^n \times \mathbb{R}^D$, $\rho > 0$, $\kappa \in (0, 1)$, and $\nu \in (0, 2\mu L/(\mu + L))$, there exist $\bar\gamma$ and $\bar\epsilon$ such that, for all $\gamma \in (0, \bar\gamma)$ and $\epsilon \in (0, \bar\epsilon)$, it holds*

$$\|x_t - \theta_\star\| \leq \left(1 - \frac{\min\{\gamma\nu, \kappa\}}{T_{max}}\right)^{\frac{t}{2}} \sqrt{T_{max}} \left\| \begin{bmatrix} x_0 - \theta_\star \\ \text{COL}(z_0^1 - \ell(x_0 + \epsilon d^1), \ldots, z_0^D - \ell(x_0 + \epsilon d^D)) \end{bmatrix} \right\| + \rho,$$

*for all $t \in \mathbb{N}$.* $\square$

The proof of Theorem 1 is provided in Section 4.4. More in detail, the proof of Theorem 1 is based on interpreting Algorithm 1 as a time-varying dynamical system and, then, on using system theory tools to characterize its evolution through some preparatory steps carried out in Section 4.

Theorem 1 ensures that for any initial conditions $(x_0, z_0)$ and desired final accuracy $\rho$, it is possible to tune Algorithm 1 to get the desired performance. From a system theory perspective, this means that $(\theta_\star, \mathrm{COL}(\theta_\star + \epsilon d^1, \ldots, \mathrm{COL}(\theta_\star + \epsilon d^D)))$ is a *semi-globally practically exponentially stable* equilibrium point (see Definition 2 in Appendix A.1) of the dynamical system describing Algorithm 1.

## 4   Analysis

In this section, we interpret Algorithm 1 as a dynamical system and, then, we analyze it through system theory tools grounded on timescale separation arguments. Indeed, in Section 4.1, we can interpret the algorithm as a time-varying two-time-scale system (see Appendix A.1), namely, the feedback interconnection between a *fast* and *slow* subsystem. Hence, we separately analyze the identified subsystems by focusing on the so-called *boundary-layer* (cf. Section 4.2) and *reduced* (cf. Section 4.3) systems, two auxiliary schemes associated to the fast and slow subsystems, respectively. Finally, in Section 4.4, we provide the proof of Theorem 1 by suitably combining the results obtained in the previous sections with timescale separation and Lyapunov-based arguments.

Assumptions 1, 2, and 3 hold true throughout the whole section.

**Remark 1.** *The key tool employed in the proof of Theorem 1 is (deterministic) timescale separation. In the case of stochastic direction selection (rather than the deterministic one imposed by Assumption 3) our proof can be adapted by relying on stochastic timescale separation (see, e.g., Carnevale and Notarstefano [2024]). In this case, the convergence results would be stated in an almost sure sense, provided that the expected value $\mathbb{E}[s_t^j]$ of each sampling selector index $s_t^j$ is strictly positive.* □

**Remark 2.** *Our proof technique based on timescale separation enables an efficient extension of the meta-algorithm analysis to more general scenarios (e.g., the nonconvex case). In particular, it promotes modularity, as it allows for modifying only the analysis of the reduced system (i.e., the gradient method, see Lemma 2) to adapt to the specific optimization setting. For the same reasons, this modularity paves the way for extending our meta-algorithm to more advanced variants using, e.g., accelerated methods Lin et al. [2020] as a baseline in place of the gradient descent method.* □

### 4.1   Two-Time-Scale System Interpretation

Let $S_t := \mathrm{diag}\{s_t^1, \ldots, s_t^D\} \in \mathbb{R}^{D \times D}$, $G_t^\epsilon : \mathbb{R}^n \times \mathbb{R}^D \to \mathbb{R}^n$, and $\mathcal{L}^\epsilon : \mathbb{R}^n \to \mathbb{R}^D$ be defined as

$$G_t^\epsilon(\theta, z) := \sum_{j=1}^D g^\epsilon(z^j + s_t^j(\ell(\theta + \gamma d^j) - z^j), d^j), \quad \mathcal{L}^\epsilon(\theta) := \begin{bmatrix} \ell(\theta + \epsilon d^1) & \ldots & \ell(\theta + \epsilon d^D) \end{bmatrix}^\top.$$

With this notation at hand, we compactly rewrite Algorithm 1 as the time-varying dynamical system

$$x_{t+1} = x_t - \gamma G_t^\epsilon(x_t, z_t) \tag{8a}$$

$$z_{t+1} = z_t + S_t(\mathcal{L}^\epsilon(x_t) - z_t). \tag{8b}$$

System (8) exhibits the following four peculiar features. First, subsystem (8b) admits an equilibrium manifold of the form $\bar{z} = \mathcal{L}^\epsilon(x_t)$, namely, that is parametrized by the other state variable $x_t$. Second, for arbitrarily fixed $x_t = x$, it can be shown that the equilibrium manifold $\bar{z} = \mathcal{L}^\epsilon(x)$ is globally exponentially stable for (8b) (see Definition 2 in Appendix A.1). Third, the variations of $x_t$ over $t$ can be made arbitrarily small by reducing the parameter $\gamma$. Fourth, in light of Assumption 2, we note that subsystem (8a), in the ideal case in which $z_t = \mathcal{L}^\epsilon(x_t)$, would approximate a gradient descent method applied to (1) with an error characterized in (2). Neglecting this error, we observe that systems with these four features are typically referred to as *two-time-scale systems* in the literature and are widely studied in system theory, see the survey Abdelgalil et al. [2023] and the dedicated Appendix A.1. The key idea is that subsystem (8a) can be made arbitrarily slow via $\gamma$, allowing the other subsystem (8b) to stay close to the current equilibrium $\mathcal{L}^\epsilon(x_t)$ and, in turn, allowing subsystem (8a) to evolve approximately as a gradient descent method. Accordingly, we refer to (8a) as the *slow subsystem*, while (8b) is termed the *fast subsystem*. To conveniently exploit this two-time-scale interpretation, we temporarily disregard the approximation error by considering an auxiliary system that serves as "nominal" version of the original system (8). Essentially, in this nominal system, we explicitly include an additive perturbation in the slow dynamics (8a) that allows for *exactly* recovering the gradient

descent method applied to problem (1) when $z_t = \mathcal{L}^\epsilon(x_t)$. Hence, this nominal system is given by

$$x_{t+1} = x_t - \gamma(G_t^\epsilon(x_t, z_t) - e_t^\epsilon(x_t)) \tag{9a}$$

$$z_{t+1} = z_t + S_t(\mathcal{L}^\epsilon(x_t) - z_t), \tag{9b}$$

where $e_t^\epsilon(x_t) := G_t^\epsilon(x_t, \mathcal{L}^\epsilon(x_t)) - \nabla\ell(x_t)$. Finally, since we want to base our analysis on Theorem 2 (cf. Appendix A.1) about generic two-time-scale systems, we rewrite (9) to shift the equilibrium point (cf. Definition 1 in Appendix A.1) of subsystem (9a) and match condition (36) in Theorem 2. To this end, we introduce the novel coordinate $\tilde{x} := x - \theta_\star \in \mathbb{R}^n$ and rewrite (9) as

$$\tilde{x}_{t+1} = \tilde{x}_t - \gamma(\tilde{G}_t^\epsilon(\tilde{x}_t, z_t) - \tilde{e}_t^\epsilon(\tilde{x}_t)) \tag{10a}$$

$$z_{t+1} = z_t + S_t(\tilde{\mathcal{L}}^\epsilon(\tilde{x}_t) - z_t), \tag{10b}$$

where $\tilde{G}_t^\epsilon : \mathbb{R}^n \times \mathbb{R}^D \to \mathbb{R}^n$, $\tilde{\mathcal{L}}^\epsilon : \mathbb{R}^n \to \mathbb{R}^D$ and $\tilde{e}_t^\epsilon : \mathbb{R}^n \to \mathbb{R}^n$ are defined as

$$\tilde{G}_t^\epsilon(\tilde{x}, z) := G_t^\epsilon(\tilde{x} + \theta_\star, z), \quad \tilde{\mathcal{L}}^\epsilon(\tilde{x}) := \mathcal{L}^\epsilon(\tilde{x} + \theta_\star), \quad \tilde{e}_t^\epsilon(\tilde{x}) := e_t^\epsilon(\tilde{x} + \theta_\star). \tag{11}$$

With this nominal system at hand, the following sections adopt the customary approach in the analysis of two-time-scale systems (see Theorem 2 in Appendix A.1). In detail, we separately study the stability of the fast dynamics (10b) and the slow one (10a) in two idealized scenarios giving rise to the so-called boundary-layer and reduced systems, respectively (cf. Sections 4.2 and 4.3). Then, in Section 4.4, we leverage these steps to establish the stability and convergence properties of the interconnected system (10), for sufficiently small values of $\gamma$, that is, when there is a sufficiently large timescale separation between the identified subsystems. Finally, we conclude the proof of Theorem 1 by showing that the original system (8) evolves closely to the nominal system (10), with a discrepancy governed by $e_t^\epsilon(x_t)$ and, thus, tunable by the parameter $\epsilon$ (cf. (2) in Assumption 2).

### 4.2 Boundary-Layer System Analysis

Now, we analyze the so-called *boundary-layer* system associated to (10), which is obtained by considering the fast dynamics (10b) with an arbitrarily fixed slow state $\tilde{x}_t = \tilde{x} \in \mathbb{R}^n$ for all $t \in \mathbb{N}$ (or, equivalently, by setting $\gamma = 0$ in (10a)). Hence, by using the error variable $\tilde{z}_t := z_t - \tilde{\mathcal{L}}^\epsilon(\tilde{x}) \in \mathbb{R}^D$ with respect to the parametrized equilibrium $\tilde{\mathcal{L}}^\epsilon(\tilde{x})$, the boundary-layer system reads as

$$\tilde{z}_{t+1} = (I_D - S_t)\tilde{z}_t. \tag{12}$$

The next lemma ensures that the origin is a globally exponentially stable equilibrium of (12).

**Lemma 1.** *Consider* (12). *Then, there exists a continuous function* $U : \mathbb{R}^D \times \mathbb{N} \to \mathbb{R}$ *such that*

$$\|\tilde{z}\|^2 \leq U(\tilde{z}, t) \leq T_{max} \|\tilde{z}\|^2 \tag{13a}$$

$$U((I_D - S_t)\tilde{z}, t+1) - U(\tilde{z}, t) \leq -\|\tilde{z}\|^2 \tag{13b}$$

$$U(\tilde{z}, t) - U(\tilde{z}', t) \leq T_{max} \|\tilde{z} - \tilde{z}'\| (\|\tilde{z}\| + \|\tilde{z}'\|), \tag{13c}$$

*for all* $\tilde{z}, \tilde{z}' \in \mathbb{R}^D$ *and* $t \in \mathbb{N}$. $\qquad\square$

The proof of Lemma 1 is provided in Appendix A.2.

### 4.3 Reduced System Analysis

In a mirrored way, we now analyze the so-called *reduced* system of the interconnection (10), which is obtained by considering the slow dynamics (10a) with the fast state in the equilibrium manifold at each iteration $t$, namely, with $z_t = \tilde{\mathcal{L}}^\epsilon(\tilde{x}_t)$ for all $t \in \mathbb{N}$. Therefore, the reduced system reads as

$$\tilde{x}_{t+1} = \tilde{x}_t - \gamma(\tilde{G}_t^\epsilon(\tilde{x}_t, \tilde{\mathcal{L}}^\epsilon(\tilde{x}_t)) - \tilde{e}_t^\epsilon(\tilde{x}_t)). \tag{14}$$

In light of the definitions of $\tilde{G}_t^\epsilon$, $\tilde{\mathcal{L}}^\epsilon$, and $\tilde{e}_t^\epsilon$ (cf. (11)), system (14) can be equivalently expressed as

$$\tilde{x}_{t+1} = \tilde{x}_t - \gamma\nabla\ell(\tilde{x}_t + \theta_\star). \tag{15}$$

Namely, the reduced system corresponds to the gradient descent method applied to problem (1). In the next lemma, we ensure that the origin is a globally exponentially stable equilibrium of (15).

**Lemma 2.** *Consider* (15). *Then, for all* $\gamma \in (0, \frac{2}{\mu + L}]$, *it holds*

$$\|\tilde{x} - \gamma\nabla\ell(\tilde{x} + \theta_\star)\|^2 - \|\tilde{x}\|^2 \leq -\gamma\frac{2\mu L}{\mu + L} \|\tilde{x}\|^2, \tag{16}$$

*for all* $\tilde{x} \in \mathbb{R}^n$. $\qquad\square$

The proof of Lemma 2 is provided in Appendix A.3.

## 4.4 Proof of Theorem 1

Now, we combine the results obtained in the previous sections to prove Theorem 1. Let us first introduce $\tilde{z} := z - \tilde{\mathcal{L}}^\epsilon(\tilde{x}) \in \mathbb{R}^D$, $\xi := \text{COL}(\tilde{x}, \tilde{z}) \in \mathbb{R}^{(n+D)}$, and compactly rewrite system (10) as

$$\xi_{t+1} = F(\xi_t, t), \tag{17}$$

where, by using a hybrid notation $\xi = \text{COL}(\tilde{x}, \tilde{z})$, $F : \mathbb{R}^{(n+D)} \times \mathbb{N} \to \mathbb{R}^{(n+D)}$, reads as

$$F(\xi, t) = \begin{bmatrix} \tilde{x} - \gamma(\tilde{G}_t^\epsilon(\tilde{x}, \tilde{z} + \tilde{\mathcal{L}}^\epsilon(\tilde{x})) - \tilde{e}_t^\epsilon(\tilde{x})) \\ (I_D - S_t)\tilde{z} - \tilde{\mathcal{L}}^\epsilon(\tilde{x} - \gamma(\tilde{G}_t^\epsilon(\tilde{x}, \tilde{z} + \tilde{\mathcal{L}}^\epsilon(\tilde{x})) - \tilde{e}_t^\epsilon(\tilde{x}))) + \tilde{\mathcal{L}}^\epsilon(\tilde{x}) \end{bmatrix}. \tag{18}$$

Analogously, by using this notation, we also rewrite the original system (8) in the compact form

$$\xi_{t+1} = F(\xi_t, t) + E_t^\epsilon(\xi_t), \tag{19}$$

in which we model the gradient estimation error term through $E_t^\epsilon : \mathbb{R}^{(n+D)} \to \mathbb{R}^{(n+D)}$ defined as

$$E_t^\epsilon(\xi) := \left[ -\gamma \tilde{e}_t^\epsilon(\tilde{x})^\top \quad \tilde{\mathcal{L}}^\epsilon(\tilde{x} - \gamma(\tilde{G}_t^\epsilon(\tilde{x}, \tilde{z} + \tilde{\mathcal{L}}^\epsilon(\tilde{x})) - \tilde{e}_t^\epsilon(\tilde{x})))^\top - \tilde{\mathcal{L}}^\epsilon(\tilde{x} - \gamma(\tilde{G}_t^\epsilon(\tilde{x}, \tilde{z} + \tilde{\mathcal{L}}^\epsilon(\tilde{x}))))^\top \right]^\top.$$

Then, given the Lyapunov function $U$ characterized in Lemma 1, we define $V : \mathbb{R}^{(n+D)} \times \mathbb{N} \to \mathbb{R}$ as

$$V(\xi, t) = U(\tilde{z}, t) + \|\tilde{x}\|^2, \tag{20}$$

where we use again $\xi = \text{COL}(\tilde{x}, \tilde{z})$. In light of (13a), for all $\xi \in \mathbb{R}^{(n+D)}$ and $t \in \mathbb{N}$, we have

$$\|\xi\|^2 \le V(\xi, t) \le T_{\max} \|\xi\|^2. \tag{21}$$

Hence, in light of (21), the level set $\Omega_c := \{\xi \in \mathbb{R}^{(n+D)} \mid V(\xi, t) \le c, \forall t \in \mathbb{N}\}$ of $V$ is compact for all $c > 0$. We want to invoke Theorem 2 (cf. Appendix A.1) about generic time-varying two-time-scale systems to characterize the increment of $V$ along the trajectories of the nominal, interconnected system (17) rather than in the auxiliary systems (12) and (15) (cf. Lemma 1 2, respectively). Theorem 2 requires (i) a Lyapunov function ensuring exponential stability of the origin for the boundary-layer system (12) (see (40)), (ii) a Lyapunov function ensuring exponential stability of the origin for the reduced system (15) (see (41)), and (iii) that the system dynamics and equilibrium function admit the bounds (37). The first point is achieved by using $U$ characterized in (13) (cf. Lemma 1). The second one follows by setting $W(\tilde{x}) = \|\tilde{x}\|^2$, using (16) (cf. Lemma 2), and noting that, being $W$ a quadratic function, it trivially satisfies (41a) and (41c). The third point is achieved since $\ell$ is continuous (cf. Assumption 1) and $e_t^\epsilon$ is bounded in compact sets (cf. Assumption 2). Hence, by Theorem 2, for all $\xi = (\tilde{x}, \tilde{z}) \in \mathbb{R}^n \times \mathbb{R}^D$, $\nu \in (0, 2\mu L/(\mu + L))$, and $\kappa \in (0, 1)$, there exists $\bar{\gamma} > 0$ such that, for all $\gamma \in (0, \bar{\gamma})$, the increment of $V$ along the trajectories of (17) satisfies

$$V(F(\xi, t), t + 1) - V(\xi, t) \le -\gamma\nu \|\tilde{x}\|^2 - \kappa \|\tilde{z}\|^2, \tag{22}$$

for all $t \in \mathbb{N}$. Moreover, the definition of $V$ (cf. (20)) and the bound (13c) lead to

$$V(\xi, t) - V(\xi', t) \le (T_{\max} + 1) \|\xi - \xi'\| (\|\xi\| + \|\xi'\|), \tag{23}$$

for all $t \in \mathbb{N}$ and $\xi, \xi' \in \mathbb{R}^{(n+D)}$. Now, we consider $\xi \in \Omega_c$ for some $c > 0$ to be defined later and add $\pm V(F(\xi, t), t + 1)$ to the increment $\Delta V(\xi, t) := V(F(\xi, t) + E_t^\epsilon(\xi), t + 1) - V(\xi, t)$ of $V$ along the trajectories of the original system (19), thus obtaining

$$\Delta V(\xi, t) = V(F(\xi, t), t + 1) - V(\xi, t) + V(F(\xi, t) + E_t^\epsilon(\xi), t + 1) - V(F(\xi, t), t + 1)$$

$$\overset{(a)}{\le} -\gamma\nu \|\tilde{x}\|^2 - \kappa \|\tilde{z}\|^2 + (T_{\max} + 1) \|E_t^\epsilon(\xi)\| (\|F(\xi, t) + E_t^\epsilon(\xi)\| + \|F(\xi, t)\|), \tag{24}$$

where in $(a)$ we use the bound (22) along the nominal trajectory and (23). Being $\Omega_c$ compact, then $\|\tilde{e}_t^\epsilon(\tilde{x})\| \le \tilde{L}$ for all $\xi = \text{COL}(\tilde{x}, \tilde{z}) \in \Omega_c$ by Assumption 2. Hence, by denoting the Lipschitz constant of $\mathcal{L}^\epsilon(\tilde{x} - \gamma(\tilde{G}_t^\epsilon(\tilde{x}, \tilde{z} + \tilde{\mathcal{L}}^\epsilon(\tilde{x})) - \tilde{e}_t^\epsilon(\tilde{x})))$ in $\Omega_c$ with $L_{\mathcal{L}^\epsilon}$ (which is finite since $\ell$ is differentiable, see Assumption 1), we have $\|E_t^\epsilon(\xi)\| \le \gamma\epsilon\tilde{L}_2$ with $\tilde{L}_2 := \tilde{L}\sqrt{1 + L_{\mathcal{L}^\epsilon}^2}$ and, thus, we can bound (24) as

$$\Delta V(\xi, t) \le -\min\{\gamma\nu, \kappa\}/T_{\max} V(\xi, t) + \epsilon\gamma(T_{\max} + 1)\tilde{L}_2(2 \|F(\xi, t)\| + \gamma\epsilon\tilde{L}_2). \tag{25}$$

By recalling the definition of $\tilde{G}_t^\epsilon$, $\tilde{\mathcal{L}}^\epsilon$, and $\tilde{e}_t^\epsilon$ (cf. (11)), and the gradient reconstruction property (2), we note that $\tilde{G}_t^\epsilon(0_n, \tilde{\mathcal{L}}^\epsilon(0_n)) - \tilde{e}_t^\epsilon(0_n) = \nabla \ell(\theta_\star) = 0_n$ since $\theta_\star$ is the optimal solution to problem (1).

By plugging this result in the definition of $F$ (cf. (18)), we get $F(0_{n+D}, t) = 0_{n+D}$ for all $t \in \mathbb{N}$. This is not surprising since $\xi = \text{COL}(x - \theta_\star, z - \mathcal{L}^\epsilon(x))$ by definition and, thus, $\xi = 0_{n+D}$ means that the solution estimate corresponds to the optimal solution $\theta_\star$, while the auxiliary variables correspond to their equilibrium values. Further, since $\tilde{\mathcal{L}}^\epsilon$ (see (11)) is continuous (cf. Assumption 1) and $E_t^\epsilon$ is bounded in compact sets (cf. Assumption 2), $F$ is Lipschitz continuous in $\xi$ within $\Omega_c$, that is

$$\|F(\xi, t) - F(\xi', t)\| \le L_F \|\xi - \xi'\|, \tag{26}$$

for all $\xi, \xi' \in \Omega_c$, $t \in \mathbb{N}$, and some finite $L_F > 0$. Thus, for all $\xi \in \Omega_c$, we can further bound (25) as

$$\Delta V(\xi, t) \le -\min\{\gamma\nu, \kappa\}/T_{\max} V(\xi, t) + \epsilon\gamma(T_{\max} + 1)\tilde{L}_2(2L_F \|\xi\| + \gamma\epsilon\tilde{L}_2). \tag{27}$$

We thus set $\gamma \in (0, \bar{\gamma})$ and, given the desired final radius $\rho \le c$ (without loss of generality), we define

$$\bar{\epsilon} := \min\left\{\min\{\gamma\nu, \kappa\}\rho^2/(T_{\max}(\gamma(T_{\max} + 1)\tilde{L}_2(2L_F\sqrt{c} + \gamma\tilde{L}_2))), 1\right\}. \tag{28}$$

Then, for all $\epsilon \in (0, \bar{\epsilon})$, we can further bound (27) as

$$\Delta V(\xi, t) \le -\min\{\gamma\nu, \kappa\}/T_{\max}\left(V(\xi, t) - \rho^2\right). \tag{29}$$

The inequality (29) ensures that the level set $\Omega_c$ is forward-invariant for system (19), i.e., $\xi_{t_0} \in \Omega_c \implies \xi_t \in \Omega_c$ for all $t \ge t_0$ along the trajectories of (19). To summarize, we recall that this property has been obtained for generic $c, \rho > 0$ and by satisfying the corresponding bounds defined by $\bar{\gamma}$ and $\bar{\epsilon}$. Therefore, by repeating all the above steps with $c > 0$ such that $\xi_0 = \text{COL}(x_0 - \theta_\star, z_0 - \mathcal{L}^\epsilon(x_0)) \in \Omega_c$, by noting that $\|x - \theta_\star\| = \|\tilde{x}\| \le \sqrt{V(\xi, t)}$ in light of the definition of $V$ (cf. (20)), and by iterating inequality (29) from $t = 0$, we ensure that the origin is a practically exponentially stable equilibrium (see Definition 2 in Appendix A.1) of system (19) and the proof concludes.

## 5  Numerical Simulations

This section numerically validates our theoretical findings with Monte Carlo simulations of $N = 20$ trials in a logistic regression scenario (cf. Section 5.1) and a ridge regression one (cf. Section 5.2).

### 5.1  Logistic Regression

In this first case, we consider $m \in \mathbb{N}$ points $p_1, \ldots, p_m \in \mathbb{R}^n$ with binary labels $l_1, \ldots, l_m \in \{-1, 1\}$, and we use them to train a linear classifier by addressing the problem

$$\min_{\theta \in \mathbb{R}^n} \quad \frac{1}{m}\sum_{k=1}^m \log\left(1 + e^{-l_k(\theta^\top p_k)}\right) + \frac{C}{2}\|\theta\|^2, \tag{30}$$

where $C > 0$ is a regularization parameter. In each trial, we randomly generate a synthetic dataset composed by $m = 1000$ points. See Appendix A.4 for details on the dataset generation. We perform two sets of comparisons. In the first one, we consider $n = 10, 25, 50$. The estimation technique we consider is inspired by extremum-seeking methods and is defined in (4). In detail, we consider $D$ directions $d^1, \ldots, d^D \in \mathbb{R}^n$ and we generate them according to $d^j = \text{COL}(\sin(\frac{\pi}{\tau^1}j + \phi^1), \ldots, \sin(\frac{\pi}{\tau^n}j + \phi^n))$ for all $j \in \{1, \ldots, D\}$, where, for each $k \in \{1, \ldots, n\}$, we set $\phi^k = \pi/2$, $\tau^k = \tau^{k-1}$ if $k$ is even and $\phi^k = 0$, $\tau^k = D \cdot 2^{((-k+1)/2)}$ if $k$ is odd. It is possible to show that Assumption 2 is satisfied with $D = 11$ when $n = 10$, $D = 29$ when $n = 25$, and $D = 53$ when $n = 50$ (see, e.g., Mimmo et al. [2024]). We run Algorithm 1 by cyclically selecting a single direction $d^j$ per iteration. Namely, for all $t \in \mathbb{N}$, we set $s_t^j = 1$ for $j = (t \mod D) + 1$ and $s_t^j = 0$ for all other directions. Thus, Assumption 3 is satisfied with $T_{\max} = D$. We consider the dimensions $n = 10, 25, 50$ and compare our scheme with the state-of-the-art 1-point schemes described in Appendix A.5, namely, with methods (57) Choi et al. [2002], (58) Chen et al. [2022], and (59) Zhang et al. [2022, 2024]. We empirically select $\epsilon = 0.1$ in all the schemes. We manually select the step sizes $\gamma$ to achieve the fastest possible convergence for each method. Table 1 reports the chosen values.

Table 1: Step size $\gamma$ used for each algorithm in 1-point (left) and 2-point (right) comparisons.

| | $n = 10$ | $n = 25$ | $n = 50$ |
|---|---|---|---|
| Algorithm 1 | 0.001 | 0.0003 | 0.0001 |
| 1-point (57) | 0.0003 | 0.0001 | 0.00004 |
| 1-point (58) | 0.003 | 0.0025 | 0.002 |
| 1-point (59) | 0.003 | 0.0025 | 0.002 |

| | $n = 50$ | $n = 200$ | $n = 300$ |
|---|---|---|---|
| Algorithm 1 | 0.02 | 0.004 | 0.002 |
| 2-point (60) | 0.02 | 0.004 | 0.002 |

Fig. 1 reports the average and 1-standard deviation band across the trials of the evolution over $t$ of the distance $\|x_t - \theta_\star\|$, where $\theta_\star \in \mathbb{R}^n$ is the optimal solution to the trial problem. Fig. 1 shows that our algorithm significantly outperforms methods (57), (58), and (59) in terms of convergence speed.

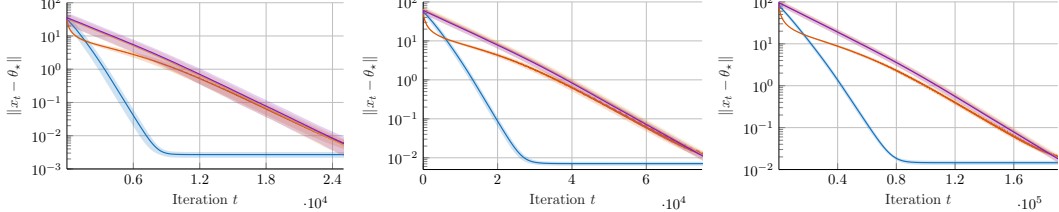

Figure 1: Problem (30) with $n = 10$ (left), $n = 25$ (middle), and $n = 50$ (right): Monte Carlo comparison of Algorithm 1 with a single cost sample per iteration (blue) and the 1-point methods (57) (red), (58) (yellow), and (59) (violet).

In the second comparison, we consider $n = 50, 200, 300$, the 2-point method (60) Agarwal et al. [2010], Duchi et al. [2015], Shamir [2017] (cf. Appendix A.5), and equip Algorithm 1 with the estimation technique (3). Namely, we consider $D = 2n$ directions $d^j$: the first $n$ correspond to the canonical basis of $\mathbb{R}^n$, while the remaining $n$ are their negatives. In this case, at each iteration $t \in \mathbb{N}$, we set $s_t^j, s_t^{j+n} = 1$ for $j = (t \mod n) + 1$ and $s_t^j = 0$ for all the other directions, so that Assumption 3 is satisfied with $T_{\max} = n$ and the comparison with (60) is fair as both methods take 2 cost samples per iteration. We empirically select $\epsilon = 0.01$ in both algorithms, while, as before, we select $\gamma$ to achieve the fastest possible convergence of both methods, and report the chosen values in Table 1. In Fig. 2, we report the average and 1-standard deviation band across the trials of the evolution over the iterations $t$ of the optimality distance $\|x_t - \theta_\star\|$ in this second set of comparisons, respectively, where $\theta_\star \in \mathbb{R}^n$ is the optimal solution to the problem of the trial. As before, Fig. 2 shows that our algorithm significantly outperforms its counterpart (60) in terms of convergence speed.

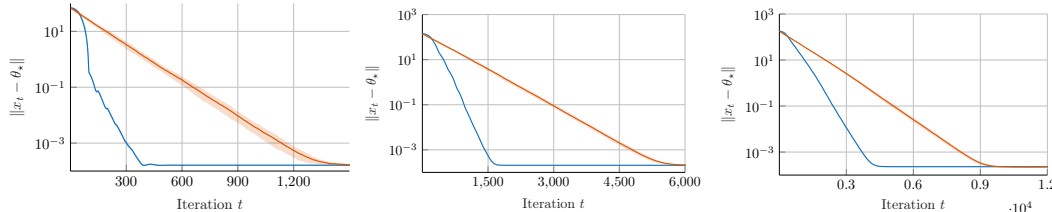

Figure 2: Problem (30): Monte Carlo comparison of Algorithm 1 with 2 cost samples per iteration (blue) and the 2-point method (60) (red) with $n = 50$ (left), $n = 200$ (middle), and $n = 300$ (right).

## 5.2  Ridge Regression

In this second case, we consider *ridge regression* problems described as

$$\min_{\theta \in \mathbb{R}^n} \quad \frac{1}{2m} \|X\theta - y\|^2 + \frac{C}{2} \|\theta\|^2, \tag{31}$$

where $X \in \mathbb{R}^{m \times n}$ is a data matrix, $y \in \mathbb{R}^m$ a response vector, and $C > 0$ is a regularization parameter. The first term enforces data fidelity, while the second term controls overfitting by promoting smoothness in the solution. See Appendix A.4 for details on the dataset generation. We consider the two sets of comparisons performed in Section 5. Hence, we first consider our method equipped with the gradient estimation technique (4) and compare it with the three 1-point methods described in Section A.5.1. The parameters of the gradient estimation technique (4) are the same used in Section 5. We empirically select $\epsilon = 0.1$ in all the algorithms. We manually select the step sizes $\gamma$ to achieve the fastest possible convergence for each method, and report the chosen values in Table 2.

Table 2: Step size $\gamma$ used for each algorithm in 1-point (left) and 2-point (right) comparisons.

| | $n = 10$ | $n = 25$ | $n = 50$ |
|---|---|---|---|
| Algorithm 1 | 0.0005 | 0.0001 | 0.00007 |
| 1-point (57) | 0.0001 | 0.00005 | 0.00002 |
| 1-point (58) | 0.001 | 0.00125 | 0.001 |
| 1-point (59) | 0.001 | 0.00125 | 0.001 |

| | $n = 50$ | $n = 200$ | $n = 300$ |
|---|---|---|---|
| Algorithm 1 | 0.01 | 0.003 | 0.002 |
| 2-point (60) | 0.01 | 0.003 | 0.002 |

In Fig. 3, we report the average and 1-standard deviation band across the trials of the evolution over the iterations $t$ of the optimality distance $\|x_t - \theta_\star\|$ in this first set of comparisons, where $\theta_\star \in \mathbb{R}^n$ is the optimal solution to the problem of the trial. Fig. 3 shows that our algorithm significantly outperforms the 1-point methods (57), (58), and (59) in terms of convergence speed.

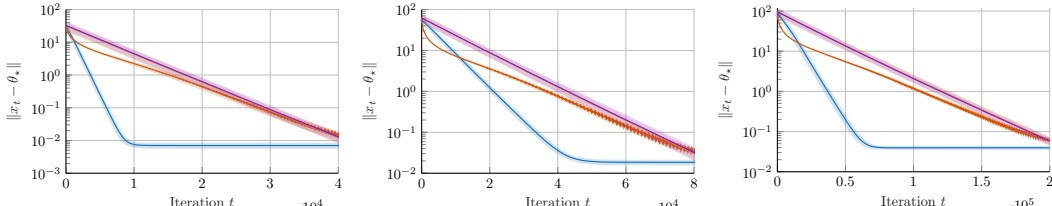

Figure 3: Problem (31) with $n = 10$ (left), $n = 25$ (middle), and $n = 50$ (right): Monte Carlo comparison of Algorithm 1 with a single cost sample per iteration (blue) and the 1-point methods (57) (red), (58) (yellow), and (59) (violet).

In the second comparison, we consider $n = 50, 200, 300$, the 2-point method (60) Agarwal et al. [2010], Duchi et al. [2015], Shamir [2017] (cf. Appendix A.5), and equip Algorithm 1 with the estimation technique (3). Namely, we consider $D = 2n$ directions $d^j$: the first $n$ correspond to the canonical basis of $\mathbb{R}^n$, while the remaining $n$ are their negatives. In this case, at each iteration $t \in \mathbb{N}$, we set $s_t^j, s_t^{j+n} = 1$ for $j = (t \mod n) + 1$ and $s_t^j = 0$ for all the other directions, so that Assumption 3 is satisfied with $T_{\max} = n$ and the comparison with (60) is fair as both methods take 2 cost samples per iteration. We empirically select $\epsilon = 0.1$ in both algorithms, while, as before, we select $\gamma$ to achieve the fastest possible convergence of both methods, and report the chosen values in Table 2. In Fig. 4, we report the average and 1-standard deviation band across the trials of the evolution over the iterations $t$ of the optimality distance $\|x_t - \theta_\star\|$ in this second set of comparisons, respectively, where $\theta_\star \in \mathbb{R}^n$ is the optimal solution to the problem of the trial. As before, Fig. 4 shows that our algorithm significantly outperforms its counterpart (60) in terms of convergence speed.

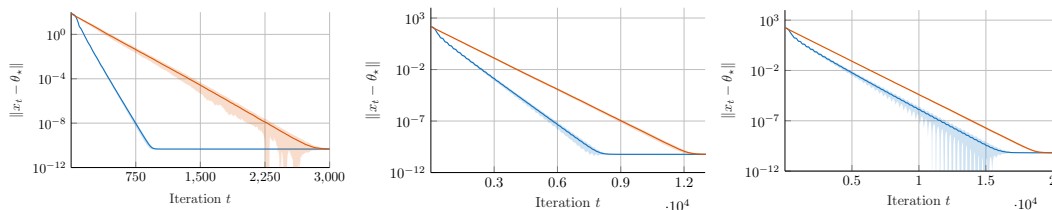

Figure 4: Problem (31): Monte Carlo comparison of Algorithm 1 with 2 cost samples per iteration (blue) and the 2-point method (60) (red) with $n = 50$ (left), $n = 200$ (middle), and $n = 300$ (right).

## 6 Conclusions

In this paper, we introduced a novel paradigm for model-free optimization based on memory mechanisms. We started from a generic technique that estimates the gradient of the cost function at a given point using cost samples obtained by perturbing that point along a finite, fixed set of directions. Our main contribution is the introduction of a memory mechanism that enables the reuse of the most recent cost samples for each direction. This mechanism removes the need to evaluate the cost in all directions at every iteration, while still allowing the estimation of descent directions through a weighted average of all directions. As a result, by using this approximate descent in the solution estimate update, the proposed approach achieves faster convergence without increasing the oracle complexity. We analyzed the algorithm using system-theoretic tools based on timescale separation and, for strongly convex problems, established convergence to an arbitrarily small neighborhood of the optimal solution. Finally, numerical experiments on regression problems demonstrated that our method significantly outperforms state-of-the-art model-free optimization algorithms.

## Acknowledgments and Disclosure of Funding

Work supported by Fondi PNRR - Bando PE - Progetto PE11 - 3A-ITALY, "Made in Italy Circolare e Sostenibile" - Codice PE0000004, CUP: J33C22002950001.

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

# A Technical Appendices and Supplementary Material

## A.1 Preliminaries on Two-Time-Scale Discrete-Time Systems

In this section, we provide a generic stability result for discrete-time two-time-scale systems. Before doing so, we introduce some preliminaries about discrete-time dynamical systems. Consider a generic time-varying discrete-time dynamical system parametrized by a parameter $\gamma > 0$ and described by

$$\chi_{t+1} = G_\gamma(\chi_t, t), \tag{32}$$

where $\chi_t \in \mathbb{R}^n$ is the state vector and $G_\gamma : \mathbb{R}^n \times \mathbb{N} \to \mathbb{R}^n$ describes its time-varying dynamics. The next definition introduces the concept of *equilibrium point*.

**Definition 1.** *A point $\bar{\chi} \in \mathbb{R}^n$ is an equilibrium point of system* (32) *if*

$$\bar{\chi} = G_\gamma(\bar{\chi}, t),$$

*for any $t \in \mathbb{N}$ and $\gamma > 0$.* □

The following definition introduces the concepts of *global and semi-global exponential (or geometric) stability* of an equilibrium point. The key distinction between the two lies in the dependence of the bound on the parameter $\gamma$ (see system (32)) with respect to the initial conditions. In the global case, there exists a bound on $\gamma$ that is uniform over all initial conditions for which stability is achieved. In contrast, in the semi-global case, for each initial condition, there exists a (possibly different) bound on $\gamma$ that ensures stability. The definition also introduces the concept of *practical stability*, which allows for an arbitrarily small bound on the asymptotic distance from a given point.

**Definition 2.** *The equilibrium point $\bar{\chi}$ is said to be globally exponentially (or geometrically) stable for* (32) *if there exists $\bar{\gamma} > 0$ such that, for all $\gamma \in (0, \bar{\gamma})$, the trajectories of* (32) *satisfy*

$$\|\chi_t - \bar{\chi}\| \le a_1 \|\chi_0 - \bar{\chi}\| a_2^t, \tag{33}$$

*for all $\chi_0 \in \mathbb{R}^n$ and some $a_1 > 0$, $a_2 \in (0, 1)$. The equilibrium point $\bar{\chi}$ is said to be semi-globally exponentially (or geometrically) stable if, for all $\chi_0 \in \mathbb{R}^n$, there exist $\bar{\gamma}, b_1, b_2 > 0$ such that, for all $\gamma \in (0, \bar{\gamma})$, the exponential decay* (33) *holds true for some $a_1 > 0$, $a_2 \in (0, 1)$. Finally, a point $\bar{\chi}_p \in \mathbb{R}^n$ is said to be semi-globally exponentially (or geometrically) practically stable if for all $\chi_0 \in \mathbb{R}^n$ and $\rho > 0$, there exist $\bar{\gamma} > 0$ such that, for all $\gamma \in (0, \bar{\gamma})$, it holds*

$$\|\chi_t - \bar{\chi}_p\| \le a_1 \|\chi_0 - \bar{\chi}_p\| a_2^t + \rho, \tag{34}$$

*for some $a_1 > 0$, $a_2 \in (0, 1)$.* □

A popular tool for establishing the stability properties of an equilibrium point is the so-called Lyapunov approach. Essentially, it consists in showing the decrease of a so-called Lyapunov function $V : \mathbb{R}^n \times \mathbb{N} \to \mathbb{R}$ along the trajectories of the system, namely, $V(G_\gamma(\chi, t), t+1) - V(\chi, t) \le 0$ for all $\chi \in \mathbb{R}^n$ and $t \in \mathbb{N}$ (see, e.g., [Haddad and Chellaboina, 2008, Ch. 13] for a detailed discussion). The following theorem applies the Lyapunov approach to establish semi-global exponential stability properties for a generic discrete-time two-time-scale system. Such a stability result extends [Carnevale et al., 2025, Th.II.3] as it weakens the assumption of global Lipschitz continuity on the subsystems' vector fields and the equilibrium function, requiring only condition (37) to hold in compact sets. As a consequence, the next theorem establishes semi-global exponential stability for the interconnected system (35), in contrast to the global result presented in [Carnevale et al., 2025, Th. II.3]. Namely, the bound on $\gamma$ under which the stability of the interconnected system is guaranteed is no longer uniform over all initial conditions (see Definition 2). To the best of the authors' knowledge, this result is novel and, thus, represents a side contribution of this paper.

**Theorem 2** (Semi-Global exponential stability for time-varying two-time-scale systems)**.** *Consider the time-varying interconnected system*

$$x_{t+1} = x_t + \gamma f(x_t, z_t, t) \tag{35a}$$

$$z_{t+1} = g(z_t, x_t, t), \tag{35b}$$

*with $x_t \in \mathcal{D} \subseteq \mathbb{R}^n$, $z_t \in \mathbb{R}^m$, $f : \mathcal{D} \times \mathbb{R}^m \times \mathbb{N} \to \mathcal{D}$, $g : \mathbb{R}^m \times \mathbb{R}^n \times \mathbb{N} \to \mathbb{R}^m$, and $\gamma > 0$. Assume that there exists $z_{eq} : \mathbb{R}^n \to \mathbb{R}^m$ such that for all $x \in \mathcal{D}$ it holds*

$$0_n = f(0_n, z_{eq}(0_n), t) \tag{36a}$$

$$z_{eq}(x) = g(z_{eq}(x), x, t), \tag{36b}$$

*for all $t \in \mathbb{N}$. Moreover, assume that for any compact set $\mathcal{S} \subset \mathcal{D} \times \mathbb{R}^m$, the constants $L_f, L_g, L_{eq} > 0$ defined as*

$$L_f := \sup_{(x,\tilde{z}) \in \mathcal{S}, t \in \mathbb{N}} \left\| \begin{bmatrix} \nabla_1 f(x, \tilde{z} + z_{eq}(x), t) \\ \nabla_2 f(x, \tilde{z} + z_{eq}(x), t) \end{bmatrix} \right\| \tag{37a}$$

$$L_g := \sup_{(x,\tilde{z}) \in \mathcal{S}, t \in \mathbb{N}} \left\| \begin{bmatrix} \nabla_1 g(\tilde{z} + z_{eq}(x), x, t) \\ \nabla_2 g(\tilde{z} + z_{eq}(x), x, t) \end{bmatrix} \right\| \tag{37b}$$

$$L_{eq} := \sup_{(x,\tilde{z}) \in \mathcal{S}, t \in \mathbb{N}} \left\| \nabla z_{eq}(x + \gamma f(x, \tilde{z} + z_{eq}(x), t)) \right\| \tag{37c}$$

*exist and are finite. Let*

$$x_{t+1} = x_t + \gamma f(x_t, z_{eq}(x_t), t) \tag{38}$$

*be the reduced system and*

$$\tilde{z}_{t+1} = g(\tilde{z}_t + z_{eq}(x), x, t) - z_{eq}(x) \tag{39}$$

*be the boundary layer system with $\tilde{z}_t \in \mathbb{R}^m$.*

*Assume that there exists a continuous function $U : \mathbb{R}^m \times \mathbb{N} \to \mathbb{R}$ such that*

$$b_1 \left\| \tilde{z} \right\|^2 \le U(\tilde{z}, t) \le b_2 \left\| \tilde{z} \right\|^2 \tag{40a}$$

$$U(g(\tilde{z} + z_{eq}(x), x, t) - z_{eq}(x), t+1) - U(\tilde{z}, t) \le -b_3 \left\| \tilde{z} \right\|^2 \tag{40b}$$

$$|U(\tilde{z}, t) - U(\tilde{z}', t)| \le b_4 \left\| \tilde{z} - \tilde{z}' \right\| (\left\| \tilde{z} \right\| + \left\| \tilde{z}' \right\|), \tag{40c}$$

*for all $\tilde{z}, \tilde{z}' \in \mathbb{R}^m$, $x \in \mathbb{R}^n$, $t \in N$, and some $b_1, b_2, b_3, b_4 > 0$. Further, assume there exist a continuous function $W : \mathcal{D} \times \mathbb{N} \to \mathbb{R}$ and $\bar{\gamma}_1 > 0$ such that, for all $\gamma \in (0, \bar{\gamma}_1)$, it holds*

$$c_1 \left\| x \right\|^2 \le W(x, t) \le c_2 \left\| x \right\|^2 \tag{41a}$$

$$W(x + \gamma f(x, z_{eq}(x), t), t+1) - W(x, t) \le -\gamma c_3 \left\| x \right\|^2 \tag{41b}$$

$$|W(x, t) - W(x', t)| \le c_4 \left\| x - x' \right\| \left\| x \right\| + c_4 \left\| x - x' \right\| \left\| x' \right\|, \tag{41c}$$

*for all $x, x' \in \mathcal{D}$, $t \in \mathbb{N}$, and some $c_1, c_2, c_3, c_4 > 0$.*

*Then, for all $(x, \tilde{z}) \in \mathcal{D} \times \mathbb{R}^m$, $\tilde{c}_3 \in (0, c_3)$, and $\tilde{b}_3 \in (0, b_3)$, there exists $\bar{\gamma} \in (0, \bar{\gamma}_1)$ such that, for all $\gamma \in (0, \bar{\gamma})$, it holds*

$$U(g(\tilde{z} + z_{eq}(x), x, t) - z_{eq}(x + \gamma f(x, \tilde{z} + z_{eq}(x), t)), t+1) + W(x + \gamma f(x, \tilde{z} + z_{eq}(x), t), t+1)$$
$$- U(\tilde{z}, t) - W(x, t) \le -\gamma \tilde{c}_3 \left\| x \right\|^2 - \tilde{b}_3 \left\| \tilde{z} \right\|^2,$$

*for all $t \in \mathbb{N}$.*

*Proof.* Let us define the error coordinate $\tilde{z}_t := z_t - z_{eq}(x_t) \in \mathbb{R}^m$ and accordingly rewrite the interconnected system (35) as

$$x_{t+1} = x_t + \gamma f(x_t, \tilde{z}_t + z_{eq}(x_t), t) \tag{42a}$$

$$\tilde{z}_{t+1} = g(\tilde{z}_t + z_{eq}(x_t), x_t, t) - z_{eq}(x_t) + \Delta z_{eq}(x_{t+1}, x_t), \tag{42b}$$

where we introduce the drift function $\Delta z_{eq}(x_{t+1}, x_t) := -z_{eq}(x_{t+1}) + z_{eq}(x_t)$. Given $c > 0$, let $\Omega_c \subset \mathcal{D} \times \mathbb{R}^m$ be the level set of the overall Lyapunov function $U(\tilde{z}, t) + W(x, t)$ (see (40) and (41)), namely

$$\Omega_c := \{(x, \tilde{z}) \in \mathcal{D} \times \mathbb{R}^m \mid U(\tilde{z}, t) + W(x, t) \le c, \forall t \in \mathbb{N}\}.$$

Now, we take a generic pair $(x, \tilde{z}) \in \Omega_c$. In light of (40a) and (41a), we note that $\Omega_c$ is compact for all $c > 0$. The compactness of $\Omega_c$, in turn, allows us to claim that, for all $t \in \mathbb{N}$, the functions $f(x, \tilde{z} + z_{eq}(x), t)$ and $g(\tilde{z} + z_{eq}(x), x, t)$ are Lipschitz continuous in their first two arguments on $\Omega_c$, and that $z_{eq}(x + \gamma f(x, \tilde{z} + z_{eq}(x), t))$ is Lipschitz continuous on $\Omega_c$, with finite constants $L_f$, $L_g$, and $L_{eq}$ defined as in (37) with $\mathcal{S} = \Omega_c$. With these constants at hand, we can follow the remaining steps in the proof of [Carnevale et al., 2025, Th.II.3] to show that the overall Lyapunov function $U(\tilde{z}, t) + W(x, t)$ is decreasing along the trajectories of (42) rather than along the trajectories of the boundary-layer and reduced systems, namely, the auxiliary dynamics (38) and (39), respectively. For the sake of completeness, we report all the steps of the proof as follows.

We now evaluate the increment $\Delta W(x, \tilde{z}, t) := W(x + \gamma f(x, \tilde{z} + z_{\text{eq}}(x), t), t + 1) - W(x, t)$ of $W(x, t)$ along the trajectories of subsystem (42a) and obtain

$$\Delta W(x, \tilde{z}, t) = W(x + \gamma f(x, \tilde{z} + z_{\text{eq}}(x), t), t + 1) - W(x, t)$$

$$\stackrel{(a)}{=} W(x + \gamma f(x, z_{\text{eq}}(x), t), t + 1) - W(x_t, t)$$
$$+ W(x + \gamma f(x, \tilde{z} + z_{\text{eq}}(x), t), t + 1) - W(x + \gamma f(x, z_{\text{eq}}(x), t), t + 1)$$

$$\stackrel{(b)}{\leq} -\gamma c_3 \|x\|^2 + W(x + \gamma f(x, \tilde{z} + z_{\text{eq}}(x), t), t + 1) - W(x + \gamma f(x, z_{\text{eq}}(x), t), t + 1)$$

$$\stackrel{(c)}{\leq} -\gamma c_3 \|x\|^2 + 2\gamma c_4 L_f \|\tilde{z}\| \|x\|$$
$$+ \gamma^2 c_4 L_f \|\tilde{z}_t\| \|f(x, \tilde{z} + z_{\text{eq}}(x), t)\| + \gamma^2 c_4 L_f \|\tilde{z}\| \|f(x, z_{\text{eq}}(x), t)\|, \tag{43}$$

where in $(a)$ we add and subtract the term $W(x + \gamma f(x, z_{\text{eq}}(x), t), t + 1)$, in $(b)$ we use (41b) to bound the difference of the first two terms, in $(c)$ we use (41c), the Lipschitz continuity of $f$, and the triangle inequality. By recalling that $f(0_n, z_{\text{eq}}(0_n), t) = 0_n$ (cf. (36)), we can write

$$\|f(x, \tilde{z} + z_{\text{eq}}(x), t)\| = \|f(x, \tilde{z} + z_{\text{eq}}(x), t) - f(0_n, z_{\text{eq}}(0_n), t)\|$$

$$\stackrel{(a)}{\leq} L_f \|x\| + L_f \|\tilde{z} + z_{\text{eq}}(x) - z_{\text{eq}}(0_n)\|$$

$$\stackrel{(b)}{\leq} L_f(1 + L_{\text{eq}}) \|x\| + L_f \|\tilde{z}\|, \tag{44}$$

where in $(a)$ we use the Lipschitz continuity of $f$ and $z_{\text{eq}}$, while in $(b)$ we combine the Lipschitz continuity of $z_{\text{eq}}$ and the triangle inequality. With similar arguments, we can show the bound

$$\|f(x, z_{\text{eq}}(x), t)\| \leq L_f(1 + L_{\text{eq}}) \|x\|. \tag{45}$$

By using the inequalities (44) and (45), we then bound the right-hand side of (43) according to

$$\Delta W(x, \tilde{z}, t) \leq -\gamma c_3 \|x\|^2 + 2\gamma c_4 L_f \|\tilde{z}\| \|x\| + \gamma^2 c_4 L_f^2 \|\tilde{z}\|^2 + 2\gamma^2 c_4 L_f^2 (1 + L_{\text{eq}}) \|\tilde{z}\| \|x\|$$

$$\stackrel{(a)}{\leq} -c_3 \|x\|^2 + \gamma^2 k_3 \|\tilde{z}\|^2 + (\gamma k_1 + \gamma^2 k_2) \|\tilde{z}\| \|x\|, \tag{46}$$

where in $(a)$ we introduce the constants $k_1, k_2, k_3 > 0$ defined as

$$k_1 := 2c_4 L_f, \quad k_2 := 2c_4 L_f^2 (1 + L_{\text{eq}}), \quad k_3 := c_4 L_f^2.$$

We now evaluate the increment $\Delta U(\tilde{z}, x, t) := U(g(\tilde{z} + z_{\text{eq}}(x), x, t) - z_{\text{eq}}(x) + \Delta z_{\text{eq}}(x + \gamma f(x, \tilde{z} + z_{\text{eq}}(x), t), x), t + 1) - U(\tilde{z}, t)$ of the function $U$ (cf. (40)) along the trajectories of (42b), obtaining

$$\Delta U(\tilde{z}, x, t) = U(g(\tilde{z} + z_{\text{eq}}(x), x, t) - z_{\text{eq}}(x) + \Delta z_{\text{eq}}(x + \gamma f(x, \tilde{z} + z_{\text{eq}}(x), t), x), t + 1) - U(\tilde{z}, t)$$

$$\stackrel{(a)}{\leq} U(g(\tilde{z} + z_{\text{eq}}(x), x, t) - z_{\text{eq}}(x), t + 1) - U(\tilde{z}, t)$$
$$- U(g(\tilde{z} + z_{\text{eq}}(x), x, t) - z_{\text{eq}}(x), t + 1)$$
$$+ U(g(\tilde{z} + z_{\text{eq}}(x), x, t) - z_{\text{eq}}(x) + \Delta z_{\text{eq}}(x + \gamma f(x, \tilde{z} + z_{\text{eq}}(x), t), x), t + 1)$$

$$\stackrel{(b)}{\leq} -b_3 \|\tilde{z}\|^2 - U(g(\tilde{z} + z_{\text{eq}}(x), x, t) - z_{\text{eq}}(x), t + 1)$$
$$+ U(g(\tilde{z} + z_{\text{eq}}(x), x, t) - z_{\text{eq}}(x) + \Delta z_{\text{eq}}(x + \gamma f(x, \tilde{z} + z_{\text{eq}}(x), t), x), t + 1)$$

$$\stackrel{(c)}{\leq} -b_3 \|\tilde{z}\|^2$$
$$+ b_4 \|\Delta z_{\text{eq}}(x + \gamma f(x, \tilde{z} + z_{\text{eq}}(x), t), x)\|$$
$$\times \|g(\tilde{z} + z_{\text{eq}}(x), x, t) - z_{\text{eq}}(x) + \Delta z_{\text{eq}}(x + \gamma f(x, \tilde{z} + z_{\text{eq}}(x), t), x)\|$$
$$+ b_4 \|\Delta z_{\text{eq}}(x + \gamma f(x, \tilde{z} + z_{\text{eq}}(x), t), x)\| \|g(\tilde{z} + z_{\text{eq}}(x), x, t) - z_{\text{eq}}(x)\|$$

$$\stackrel{(d)}{\leq} -b_3 \|\tilde{z}\|^2 + b_4 \|\Delta z_{\text{eq}}(x + \gamma f(x, \tilde{z} + z_{\text{eq}}(x), t), x)\|^2$$
$$+ 2b_4 \|\Delta z_{\text{eq}}(x + \gamma f(x, \tilde{z} + z_{\text{eq}}(x), t), x)\| \|g(\tilde{z} + z_{\text{eq}}(x), x, t) - z_{\text{eq}}(x)\|, \tag{47}$$

where in $(a)$ we add and subtract the term $U(g(\tilde{z} + z_{\text{eq}}(x), x, t) - z_{\text{eq}}(x), t + 1)$, in $(b)$ we exploit (40b) to bound the first two terms, in $(c)$ we use (40c) to bound the difference of the last two terms, and

in $(d)$ we use the triangle inequality. The definition of $\Delta z_{\text{eq}}(x + \gamma f(x, \tilde{z} + z_{\text{eq}}(x), t), x)$ and the Lipschitz continuity of $z_{\text{eq}}$ lead to

$$
\begin{aligned}
\|\Delta z_{\text{eq}}(x + \gamma f(x, \tilde{z} + z_{\text{eq}}(x), t), x)\| &\leq L_{\text{eq}} \|x + \gamma f(x, \tilde{z} + z_{\text{eq}}(x), t) - x\| \\
&\stackrel{(a)}{\leq} \gamma L_{\text{eq}} \|f(x, \tilde{z} + z_{\text{eq}}(x), t)\| \\
&\stackrel{(b)}{\leq} \gamma L_{\text{eq}} \|f(x, \tilde{z} + z_{\text{eq}}(x), t) - f(0_n, z_{\text{eq}}(0_n), t)\| \\
&\stackrel{(c)}{\leq} \gamma L_{\text{eq}} L_f (1 + L_{\text{eq}}) \|x\| + \gamma L_{\text{eq}} L_f \|\tilde{z}\|,
\end{aligned}
\tag{48}
$$

where in $(a)$ we use the update (42a), in $(b)$ we add the term $f(0_n, z_{\text{eq}}(0_n), t)$ since this is zero in light of (36), and in $(c)$ we use the triangle inequality and the Lipschitz continuity of the functions $f$ and $z_{\text{eq}}$. Moreover, since $g(z_{\text{eq}}(x), x, t) = z_{\text{eq}}(x)$ for all $x \in \mathcal{D}$ (cf. (36)), we can write

$$
\begin{aligned}
\|g(\tilde{z} + z_{\text{eq}}(x), x, t) - z_{\text{eq}}(x)\| &= \|g(\tilde{z} + z_{\text{eq}}(x), x, t) - g(z_{\text{eq}}(x), x, t)\| \\
&\leq L_g \|\tilde{z}\|,
\end{aligned}
\tag{49}
$$

where the inequality is due to the Lipschitz continuity of the function $g$. By using the inequalities (48) and (49), we then bound (47) as

$$
\begin{aligned}
\Delta U(\tilde{z}, x, t) &\leq -b_3 \|\tilde{z}\|^2 + 2\gamma b_4 L_{\text{eq}} L_g L_f (1 + L_{\text{eq}}) \|x\| \|\tilde{z}\| + 2\gamma b_4 L_{\text{eq}} L_g L_f \|\tilde{z}\|^2 \\
&\quad + \gamma^2 b_4 L_{\text{eq}}^2 L_f^2 (1 + L_{\text{eq}})^2 \|x\|^2 + 2\gamma^2 b_4 L_{\text{eq}}^2 L_f^2 (1 + L_{\text{eq}}) \|x\| \|\tilde{z}\| \\
&\quad + \gamma^2 b_4 L_{\text{eq}}^2 L_f^2 \|\tilde{z}\|^2 \\
&\leq (-b_3 + \gamma k_6 + \gamma^2 k_7) \|\tilde{z}\|^2 + \gamma^2 k_8 \|x\|^2 + (\gamma k_4 + \gamma^2 k_5) \|x\| \|\tilde{z}\|,
\end{aligned}
\tag{50}
$$

where we introduce the constants

$$
\begin{aligned}
k_4 &:= 2b_4 L_{\text{eq}} L_g L_f (1 + L_{\text{eq}}), & k_5 &:= 2b_4 L_{\text{eq}}^2 L_f^2 (1 + L_{\text{eq}}), \\
k_6 &:= 2b_4 L_{\text{eq}} L_g L_f, & k_7 &:= b_4 L_{\text{eq}}^2 L_f^2, \\
k_8 &:= b_4 L_{\text{eq}}^2 L_f^2 (1 + L_{\text{eq}})^2.
\end{aligned}
$$

We now introduce the overall Lyapunov function $V : \mathcal{D} \times \mathbb{R}^m \times \mathbb{N} \to \mathbb{R}$ defined as

$$
V(x, \tilde{z}, t) := W(x, t) + U(\tilde{z}, t).
$$

By evaluating the increment $\Delta V(x, \tilde{z}, t) := V(x + \gamma f(x, \tilde{z} + z_{\text{eq}}(x), t), g(\tilde{z} + z_{\text{eq}}(x), x, t) - z_{\text{eq}}(x) + \Delta z_{\text{eq}}(x + \gamma f(x, \tilde{z} + z_{\text{eq}}(x), t), x), t + 1) - V(x, \tilde{z}, t) = \Delta W(x, \tilde{z}, t) + \Delta U(\tilde{z}, x, t)$ of $V$ along the trajectories of the interconnected system (42), we can use the results (46) and (50) to write

$$
\Delta V(x, \tilde{z}, t) \leq - \begin{bmatrix} \|x\| \\ \|\tilde{z}\| \end{bmatrix}^\top Q(\gamma) \begin{bmatrix} \|x\| \\ \|\tilde{z}\| \end{bmatrix},
\tag{51}
$$

where we introduce the matrix $\mathcal{Q}(\gamma) = \mathcal{Q}(\gamma)^\top \in \mathbb{R}^2$ defined as

$$
\mathcal{Q}(\gamma) := \begin{bmatrix} \gamma c_3 - \gamma^2 k_8 & q_{21}(\gamma) \\ q_{21}(\gamma) & b_3 - \gamma k_6 - \gamma^2 (k_3 + k_7) \end{bmatrix},
$$

with $q_{21}(\gamma) := -\frac{1}{2}(\gamma(k_1 + k_4) + \gamma^2(k_2 + k_5))$. Now, we consider the constants $\tilde{c}_3, \tilde{b}_3$ introduced in the theorem statement and we recall that $\tilde{c}_3 \in (0, c_3)$ and $\tilde{b}_3 \in (0, b_3)$. By Sylvester criterion, we know that $\mathcal{Q} > \begin{bmatrix} \gamma \tilde{c}_3 & 0 \\ 0 & \tilde{b}_3 \end{bmatrix}$ if and only if

$$
\begin{cases} \gamma(c_3 - \tilde{c}_3) > \gamma^2 k_8 \\ \gamma(c_3 - \tilde{c}_3)(b_3 - \tilde{b}_3) > p(\gamma), \end{cases}
\tag{52}
$$

where the polynomial $p(\gamma)$ is defined as

$$
p(\gamma) := q_{21}\gamma^2 + \gamma^2(c_3 - \tilde{c}_3)k_6 + \gamma^3(c_3 - \tilde{c}_3)(k_3 + k_7) + \gamma^2(b_3 - \tilde{b}_3)k_8 - \gamma^3 k_6 k_8 - \gamma^4 k_8(k_3 + k_7).
$$

First, we note that the first condition in (52) is satisfied for all $\gamma \in (0, (c_3 - \tilde{c}_3)/k_8)$. Then, we note that $p$ is a continuous function of $\gamma$ and $\lim_{\gamma \to 0} p(\gamma)/\gamma = 0$. Hence, there exists some $\bar{\gamma} \in (0, \bar{\gamma}_1)$ such that (52) is satisfied for all $\gamma \in (0, \bar{\gamma})$ and the proof concludes. $\qquad \square$

## A.2 Proof of Lemma 1

Consider the linear, time-varying system (12). Given $t_0, t \in \mathbb{N}$, we introduce the so-called state transition matrix $\Phi(t, t_0) \in \mathbb{R}^{D \times D}$ of system (12) defined as

$$\Phi(t, t_0) := \prod_{\tau=t_0}^{t-1} (I_D - S_\tau). \tag{53}$$

We note that this transition matrix allows us to write $\tilde{z}_t = \phi(t, t_0)\tilde{z}_{t_0}$ along the trajectories of system (12). Moreover, in light of the essentially-cyclic condition ensured by Assumption 3, we note that

$$\Phi(t_0 + T_{\max} + \tau, t_0) = 0_{D \times D}, \tag{54}$$

for all $t_0, \tau \in \mathbb{N}$, where $0_{D \times D}$ denotes the zero matrix of dimension $D \times D$. Then, we use $\Phi$ to explicitly define the (time-varying) Lyapunov function $U : \mathbb{R}^D \times \mathbb{N} \to \mathbb{R}$ as

$$U(\tilde{z}, t) := \sum_{\tau=t}^{t+T_{\max}-1} \|\Phi(\tau, t)\tilde{z}\|^2. \tag{55}$$

Being the chosen function quadratic, it trivially satisfies the conditions (13a) and (13c). As for condition (13b), by definition (55), we note that

$$U((I_D - S_t)\tilde{z}, t+1) = \sum_{\tau=t+1}^{t+T_{\max}} \|\Phi(\tau, t+1)(I_D - S_t)\tilde{z}\|^2$$

$$\overset{(a)}{=} \|\Phi(t+T_{\max}, t+1)(I_D - S_t)\tilde{z}\|^2 + \sum_{\tau=t+1}^{t+T_{\max}-1} \|\Phi(\tau, t+1)(I_D - S_t)\tilde{z}\|^2$$

$$\overset{(b)}{=} \|\Phi(t+T_{\max}, t)\tilde{z}\|^2 + \sum_{\tau=t+1}^{t+T_{\max}-1} \|\Phi(\tau, t)\tilde{z}\|^2$$

$$\overset{(c)}{=} U(\tilde{z}, t) - \|\tilde{z}\|^2,$$

where in $(a)$ we isolate the last term of the sum, in $(b)$ we use the fact that $\Phi(\tau, t+1)(I_D - S_t) = \Phi(\tau, t)$ for all $\tau \geq t + 1$ by definiton (53), in $(c)$ we use (54) to cancel out the first term and the definiton of $U$ (cf. (55)) to manipulate the second term as $\sum_{\tau=t+1}^{t+T_{\max}-1} \|\Phi(\tau, t)\tilde{z}\|^2 = \sum_{\tau=t}^{t+T_{\max}-1} \|\Phi(\tau, t)\tilde{z}\|^2 - \|\Phi(t, t)\tilde{z}\|^2 = U(\tilde{z}, t) - \|\tilde{z}\|^2$ and the proof concludes.

## A.3 Proof of Lemma 2

The proof is straightforward since the reduced system (15) is the gradient method applied to problem (1). We report it here for completeness. We consider the increment $\|\tilde{x} - \gamma \nabla \ell(\tilde{x} + \theta_\star)\|^2 - \|\tilde{x}\|^2$ and, by expanding the square norm, we obtain

$$\|\tilde{x} - \gamma \nabla \ell(\tilde{x} + \theta_\star)\|^2 - \|\tilde{x}\|^2$$

$$= -2\gamma \nabla \ell(\tilde{x} + \theta_\star)^\top \tilde{x} + \gamma^2 \|\nabla \ell(\tilde{x} + \theta_\star)\|^2$$

$$\overset{(a)}{\leq} -2\gamma \left(\nabla \ell(\tilde{x} + \theta_\star) - \nabla \ell(\theta_\star)\right)^\top (\tilde{x} + \theta_\star - \theta_\star) + \gamma^2 \|\nabla \ell(\tilde{x} + \theta_\star) - \nabla \ell(\theta_\star)\|^2$$

$$\overset{(b)}{\leq} -\gamma \frac{2\mu L}{\mu + L} \|\tilde{x}\|^2 - \gamma \left(\frac{2}{\mu + L} - \gamma\right) \|\nabla \ell(\tilde{x} + \theta_\star) - \nabla \ell(\theta_\star)\|^2, \tag{56}$$

where in $(a)$ we use the fact that $\nabla \ell(\theta_\star) = 0_n$, while in $(b)$ we use the fact that $\ell$ is $\mu$-strongly convex and its gradient is $L$-Lipschitz continuous (cf. Assumption 1). The proof concludes by using the condition $\gamma \in (0, \frac{2}{\mu+L}]$ to neglect the second term in (56).

## A.4 Dataset Generation for the Tests in Section 5

In this section, we describe how we generate the datasets used in the numerical simulations presented in Section 5.

### A.4.1 Logistic Regression Dataset Generation

In the logistic regression scenario (cf. Section 5.1), for each trial and $k \in \{1, \ldots, m\}$, we generate an input vector $p_k \in \mathbb{R}^n$ by sampling it from a standard normal distribution in $\mathbb{R}^n$ and we assign it the label $l_k = \mathrm{sign}(\bar{\theta}^\top p_k + \eta_k)$, where the ground truth vector $\bar{\theta} \in \mathbb{R}^n$ is sampled from a standard normal distribution in $\mathbb{R}^n$, $\eta_k$ is a zero-mean Gaussian noise with standard deviation $0.1$, while $\mathrm{sign} : \mathbb{R} \to \{-1, 1\}$ denotes the sign function, i.e., $\mathrm{sign}(\eta) = 1$ if $\eta \geq 0$ and $\mathrm{sign}(\eta) = -1$ otherwise. Finally, we set $C = 1$ and, in each trial, we randomly initialize the algorithms' solution estimates by sampling $x_0$ from a zero-mean Gaussian distribution with covariance matrix $10 \cdot I_n$.

### A.4.2 Ridge Regression Dataset Generation

In the ridge regression scenario (cf. Section 5.2), in each Monte Carlo trial, the entries of $X$ are sampled from a standard Gaussian distribution, while the ground truth vector $\bar{\theta} \in \mathbb{R}^n$ is drawn from a standard Gaussian distribution in $\mathbb{R}^n$. The response vector $y$ is then constructed as

$$y = X\bar{\theta} + \eta,$$

where $\eta \in \mathbb{R}^m$ is randomly generated by sampling it from a zero-mean Gaussian distribution with standard deviation $0.1$. Finally, we set $C = 1$ and, in each trial, we randomly initialize the algorithms' solution estimates by sampling $x_0$ from a zero-mean Gaussian distribution with covariance matrix $10 \cdot I_n$.

## A.5 Description of the Algorithms Used in the Comparisons in Section 5

In this section, we provide the description of the algorithms used in the comparisons with our method performed in Section 5. We recall that the choice of the parameters $\gamma$ and $\epsilon$ for these methods is provided in Section 5 and Appendix 5.2.

### A.5.1 1-point methods

We consider three 1-point methods. The first one is a discrete-time extremum-seeking method (see, e.g., Choi et al. [2002]) and consists in maintaining a solution estimate $x_t \in \mathbb{R}^n$ and a low-pass filter $z_t \in \mathbb{R}$ and updating them according to

$$x_{t+1} = x_t - \gamma \frac{2(\ell(x_t + \epsilon d_t) - z_t)d_t}{\epsilon} \tag{57a}$$

$$z_{t+1} = z_t + \gamma(\ell(x_t + \epsilon d_t) - z_t), \tag{57b}$$

where $d_t \in \mathbb{R}^n$ is a sinusoidal signal, namely, $d_t = \mathrm{COL}(\sin(\frac{\pi}{\tau^1}t + \phi^1), \ldots, \sin(\frac{\pi}{\tau^n}t + \phi^n))$, where $\tau^i > 0$ and $\phi^i \in [0, 2\pi)$ for all $k \in \{1, \ldots, n\}$. In the simulations, all the parameters $\tau^i$ and $\phi^i$ are chosen as in our method (see Section 5). Second, we consider the method proposed in Chen et al. [2022] that consists in maintaining a solution estimate $x_t \in \mathbb{R}^n$ and a filter variable $z_t \in \mathbb{R}$ and updating them according to

$$z_{t+1} = (1 - \beta)z_t + \ell(x_t + \epsilon d_t) - \ell(x_{t-1} + \epsilon d_{t-1}) \tag{58a}$$

$$x_{t+1} = x_t - \gamma \frac{z_{t+1}d_t}{\epsilon}, \tag{58b}$$

where $\{d_t\}_{t \in \mathbb{N}}$ are i.i.d. random directions uniformly sampled from the sphere in $\mathbb{R}^n$ with unitary radius, while $\beta > 0$ is an additional tuning parameters. In all the simulations, we run (58) by empirically setting $\beta = 0.9$. Finally, we consider the 1-point method proposed in Zhang et al. [2022, 2024] and consists in maintaining a solution estimate $x_t \in \mathbb{R}^n$ and updating it according to

$$x_{t+1} = x_t - \gamma \frac{(\ell(x_t + \epsilon d_t) - \ell(x_{t-1} + \epsilon d_{t-1}))d_t}{\epsilon}, \tag{59a}$$

where, also in this case, $\{d_t\}_{t \in \mathbb{N}}$ are i.i.d. random directions uniformly sampled from the sphere in $\mathbb{R}^n$ with unitary radius.

### A.5.2 2-point method

We consider the 2-point method proposed in Agarwal et al. [2010], Duchi et al. [2015], Shamir [2017] that consists in maintaining a solution estimate $x_t \in \mathbb{R}^n$ and updating it according to

$$x_{t+1} = x_t - \gamma \frac{(\ell(x_t + \epsilon d_t) - \ell(x_t - \epsilon d_t))d_t}{2\epsilon}, \tag{60}$$

where $\{d_t\}_{t \in \mathbb{N}}$ are i.i.d. random directions uniformly sampled from the sphere in $\mathbb{R}^n$ with unitary radius.

