# OpenReview forum: "Accelerating Model-Free Optimization via Averaging of Cost Samples"
_NeurIPS.cc/2025/Conference — NeurIPS 2025 poster_

### Official Review · Reviewer_KNsy · 2025-06-22

**Clarity:** 2
**Significance:** 3
**Originality:** 3
**Rating:** 4
**Confidence:** 2

**Summary:**

This paper proposes a simple memory mechanism for single-sample zeroth-order optimization. Instead of discarding past cost evaluations, the algorithm maintains a buffer that stores the most recent cost of each predefined perturbation direction. Under strong convexity, the method converges at a linear rate to a neighborhood of the optimal solution.

**Questions:**

1. Can the convergence analysis be generalized to weaker conditions, for example, smooth (but not strongly) convex objectives or stochastic noise in function values?

2. Would the authors consider adding stronger baselines and real-world benchmarks? In addition, given that the synthetic experiment uses a modest dimension (n=50), might larger-scale experiments be included to illustrate scalability?

3. Have you explored a principled or adaptive strategy for selecting the parameters (e.g., gamma and epsilon)?

**Ethical Concerns:**

["NO or VERY MINOR ethics concerns only"]

**Final Justification:**

The paper presents a simple yet novel memory mechanism for single-point ZO methods with rigorous convergence analysis. The rebuttal addressed concerns such as limited baselines and the absence of larger-scale experiments. Overall, the strengths and additional evidence outweigh the remaining limitations, and I recommend a positive score.

**Limitations:**

yes

**Quality:**

2

**Strengths And Weaknesses:**

### Strengths
1. Appending a simple memory mechanism to a classical single-point ZO method is both novel and easy to implement.

2. Convergence is rigorously established, with explicit rates provided.


### Weaknesses
1. Because the analysis relies on strong convexity and noiseless evaluations, the current theory may not cover many general convex problems.

2. Experiments focus on a 50-dimensional synthetic logistic-regression task and compare against only two baseline methods. Additional studies on larger dimensions, real-world black-box tasks, or stronger baselines would better demonstrate practical advantages.

3. The paper reports that parameters (e.g., step size gamma) were tuned manually, which may introduce bias into the comparisons.

---

> ### Author Rebuttal · Authors · 2025-07-29
>
> We are very happy that you appreciated the idea characterizing our method and that you found our convergence analysis rigorous. We thank for all your useful questions and observations which allowed us to improve the quality of our paper. In the following, we provide a detailed account of the changes we would make if the paper is accepted, as well as those we have already included in the revised version.
>
>
> We thank you for this observation about the fact that our analysis needs strongly convex cost functions. The main scope of the paper is the proposal of our memory paradigm and, thus, we focused on the strongly convex case for this first step. However, the modularity of our line of proof based on timescale separation allows for efficiently extending the analysis to more general scenarios as it allows for only modifying the analysis of the reduced system (i.e., the gradient method) in the considered setting (e.g., the one with merely convex functions) without modifying the other steps of the analysis. If the paper is accepted, we will add a remark to discuss this aspect.
>
>
> We thank you for your suggestions on how to improve our experimental evaluation. In response, we have already added two additional state-of-the-art 1-point algorithms for comparison, bringing the total number of competitors to four. Moreover, we introduced new experiments on ridge regression problems (see Appendix A.6 in the revised version of the paper). Finally, for all scenarios, we included tests on both larger problems (with dimensions $n = 200$ and $n = 300$) and smaller ones (with $n = 10$ and $n = 25$).
>
> We thank you for raising the point about potential bias due to manual tuning. We manually tuned the parameters to get the best performance for each method in the sense that we tested the performance of each method for a range of values of $\gamma$ and selected the one that achieved the best performance. This tuning procedure strongly mitigates potential biases in the comparisons.
>
> Thank you for your question regarding stochastic noise. We highlight that our theoretical result is grounded in a stability analysis of the proposed algorithm. As such, preliminary robustness guarantees are inherently provided by the robustness properties associated with stability. Moreover, our system-theoretic proof lays the groundwork for deriving more detailed robustness guarantees by leveraging well-established tools from the robustness theory of dynamical systems. If the paper is accepted, we will add some insights on this aspect.
>
> We thank you for the question about adaptive parameters. Studying algorithms that adaptively update their parameters is indeed a fascinating research direction and certainly worth exploring. However, due to its complexity and the fact that it remains a relatively open field, it lies beyond the scope of the present work.
>
> Thank you again for your patience and effort in reviewing our paper!

---

### Official Review · Reviewer_VWjt · 2025-06-27

**Clarity:** 3
**Significance:** 2
**Originality:** 3
**Rating:** 4
**Confidence:** 3

**Summary:**

This paper introduces a memory-based paradigm for model-free (derivative-free) optimization. The key innovation is an auxiliary memory mechanism that stores function evaluations for each perturbation direction and reuses them across iterations. The algorithm maintains memory variables that are updated when new function evaluations become available and uses weighted averages of all stored samples to estimate descent directions. The authors also provide theoretical proof of this method, then use numerical experiments on logistic regression demonstrate its performance.

**Questions:**

1. The experiments are limited to 53-dimensional logistic regression. Could the authors provide results on different dimensional case?

2. In chapter 5, the author mentioned γ is manually selected, and also different for two method. It would be better to show how different γ affect the result and compare the results with same γ. Could the authors provide sensitivity analysis of performance to parameter choices

3. Since the method needs to store a vector, can you discuss how memory efficiency and computational cost in your method?

**Ethical Concerns:**

["NO or VERY MINOR ethics concerns only"]

**Final Justification:**

I will keep my score after discussion with authors and other reviewers.

**Limitations:**

No. I think that the sensitivity to parameter choices is limited in discussion.

**Quality:**

3

**Strengths And Weaknesses:**

The memory mechanism is intuitive and makes sense. This paper also use the two-timescale system framework to provide rigorous theoretical analysis.

The idea of using historical information is well-established in optimization (momentum, Adam, etc.), though the application to zeroth-order methods is novel.

This method requires Lipschitz continuous gradients and strong convexity, limiting applicability to broader function classes.
And the numerical experiment only tested on logistic regression, lacking evaluation on other problem types also didn't compare with SOTA derivative-free optimization algorithms.

---

> ### Author Rebuttal · Authors · 2025-07-29
>
> We are happy that you appreciated our memory mechanism as well as our theoretical analysis and we thank you for recognizing the novelty of our idea in this context.  We thank you for your useful suggestions. In the following, we provide a detailed account of the changes we would make if the paper is accepted, as well as those we have already included in the revised version.
>
>
> We thank you for your comment about the applicability of our analysis. The main scope of the paper is the proposal of our memory paradigm and, thus, we focused on the strongly convex case for this first step. However, the modularity of our line of proof based on timescale separation allows for efficiently extending the analysis to more general scenarios as it allows for only modifying the analysis of the reduced system (i.e., the gradient method) in the considered setting without modifying the other steps in the proof. If the paper is accepted, we will explicitly discuss this aspect in the revised version of the paper.
>
> We thank you for all your useful suggestions to improve our experiments. In response, we already improved this aspect by taking three actions. First, we have added new experiments on ridge regression scenarios. Second, we have added comparisons with two additional SOTA 1-point algorithms, bringing the total number of algorithms considered in the comparisons to four. Finally, in all scenarios, we added experiments with larger problem dimensions $n = 200$ and $n = 300$, as well as problem with smaller dimensions $n = 10$ and $n = 25$.
>
> We thank you for the observation about the tuning of $\gamma$. However, we highlight that we manually tuned the parameters to get the best performance for each method. Using the same $\gamma$ for all methods would not be fair as it would not allow each method to achieve its best performance.
>
>
> We thank you for the question about the memory efficiency and computational efficiency of our method. Our method requires storing a $D$-dimensional vector compared to standard algorithms in this field (see below Algorithm 1). As for the computational complexity, we highlight that our method improves the convergence rate without increasing the oracle complexity. If the paper is accepted, we will discuss these aspects in the revised version of the paper.
>
>
> Thank you again for your patience and effort in reviewing our paper!

---

> ### Comment · Reviewer_VWjt · 2025-08-02
> **keep the score**
>
> Thank you for the rebuttal and I will keep my score. I thank the authors for the response and clarification of the issues raised.

---

### Official Review · Reviewer_p5uW · 2025-06-27

**Clarity:** 3
**Significance:** 3
**Originality:** 2
**Rating:** 4
**Confidence:** 4

**Summary:**

This paper introduces a meta-algorithm for zeroth-order optimization designed to accelerate convergence without increasing the number of function queries per iteration. The key idea is to challenge the standard practice of discarding cost function samples from previous iterations. The proposed method maintains an auxiliary vector of stored cost samples, one for each of a fixed set of $D$ perturbation directions. When a cost is evaluated for a particular direction, its corresponding entry in the memory vector is updated; otherwise, the old value is retained. The descent direction at each step is then calculated by averaging all $D$ perturbation directions, weighted by their corresponding cost samples from the memory vector.

The authors provide a rigorous theoretical analysis for the strongly convex case by interpreting the algorithm as a time-varying two-time-scale dynamical system. They prove that the method achieves a linear convergence rate to an arbitrarily small neighborhood of the optimal solution.

**Questions:**

Thank you for this interesting and well-written paper. I have a few questions to better understand the nuances of your work:

1. On the $T_{\max}$ Trade-off: The convergence rate in Theorem 1 includes the term $\min{\gamma \mu, \kappa}/T_{\max}$. This seems to imply that a larger $T_{\max}$ (resulting from using more directions $D$ with single-sample-per-iteration cyclic updates) would slow down the linear convergence factor. This appears to be in tension with the need for a sufficiently rich set of directions to accurately approximate the gradient. Could you comment on this trade-off? Is there an optimal number of directions $D$ that balances the quality of the gradient estimate against the staleness of the cost samples?

2. Stochastic Sampling Analysis: You mention that Assumption 3 could be extended to a stochastic setting. Could you elaborate on the main technical challenges in doing so? In your two-time-scale analysis, the fast-layer dynamics for $z$ converge due to the guaranteed periodic updates. If the selector $s_\tau^j$ becomes a random variable, how would you ensure the fast subsystem $z_t$ remains sufficiently close to the equilibrium $L_\epsilon(x_t)$ for the slow dynamics to be well-approximated by the reduced system?

3. Comparison to Momentum: How do you see your memory mechanism relating to momentum-based methods in ZO optimization? Momentum methods also leverage past information by maintaining a velocity vector, which is an exponential moving average of past gradient estimates. Your method, in contrast, stores raw cost samples. What are the conceptual advantages of storing costs over storing aggregated gradient estimates?

**Ethical Concerns:**

["NO or VERY MINOR ethics concerns only"]

**Final Justification:**

Based on the discussion with authors, I have suitably updated my score. They seem to have addressed most of my concerns. The generality of the results is the only limiting factor in my opinion.

**Limitations:**

I can think of a few limitations:

1. Reliance on Strong Convexity and Deterministic Sampling: The theoretical analysis (Theorem 1) relies on two strong assumptions. First, the objective function $l$ is assumed to be $\mu$-strongly convex (Assumption 1). This is crucial for establishing the stability of the reduced system (Lemma 2) and ensuring that the gradient descent dynamics converge to a unique optimum. Extending the analysis to the general convex or non-convex case is a non-trivial but important next step. For non-convex functions, it is unclear if the method would be guaranteed to converge to a stationary point or how the memory mechanism would interact with local minima and saddle points. Of course, lack of strong convexity will probably not guarantee linear convergence, which is expected.

Second, the proof of linear convergence hinges on the deterministic, essentially-cyclic sampling condition (Assumption 3), which guarantees that every direction is sampled within a finite window $T_{\max}$. This is essential for the stability of the boundary-layer system (Lemma 1), as it ensures that no component of the memory vector $z_t$ can remain stale indefinitely. Many popular ZO methods employ random sampling of perturbation directions. Adapting the analysis to such a stochastic setting would require different analytical tools, likely establishing convergence in expectation or with high probability.

2. Memory and Dimensionality Trade-off: The core of the method is a memory vector $z$ of size $D$, where $D$ is the number of perturbation directions. This introduces a memory cost of $O(D)$ that is not present in standard single-point methods. For many gradient estimation techniques, $D$ scales with the problem dimension $n$ (e.g., $D=2n$ for a two-sided gradient estimator). While this is acceptable for moderately-sized problems, the memory overhead could become a practical limitation in very high-dimensional settings where $n$ is in the millions. This presents a clear trade-off: the algorithm trades $O(D)$ memory for faster convergence.

**Quality:**

3

**Strengths And Weaknesses:**

Strengths:

1. The idea of maintaining and reusing the most recent cost sample for each perturbation direction is simple, intuitive, and, to the best of my knowledge, novel in this context.

2. The convergence analysis is a major strength. Modeling the algorithm as a two-time-scale system is an appropriate choice. The resulting guarantee of linear convergence to a tunable neighborhood (Theorem 1) is a strong theoretical result for this class of algorithms. The inclusion of the detailed derivation in the appendix (especially Theorem 2) is a valuable contribution in its own right.

3. The algorithm's main promise is faster convergence without increasing function queries. The method appears easy to implement and adds only a small memory overhead ($D$ scalars).

Weaknesses:

1. The core idea is not too different from coordinate descent, where descent is performed in one direction at a time. In fact, a very recent work by Karandikar et al. (Revisiting Stochastic Approximation and Stochastic Gradient Descent) has strong parallels with the proposed work in a more general setting, where they also derive sufficient conditions for the convergence of zero-order SGD, wherein the stochastic gradient is computed using $2D$ function evaluations, but no gradient computations. As such the core idea does not appear to be novel.

2. Assumption 3 introduces $T_{\max}$, the maximum time between sampling any given direction. The convergence rate in Theorem 1 depends on $1/T_{\max}$. This suggests that using more directions $D$ (which would likely increase $T_{\max}$ under cyclic sampling) could lead to a slower convergence rate. This creates an interesting trade-off: more directions might give a better gradient estimate, but the staleness of information could slow convergence. A more explicit discussion of this trade-off would improve the paper. Also, convergence of algorithms under assumptions of a constant such as $T_{\max}$ is widely dealt in the distributed optimization literature wherein the network is assumed to be connected over any $T_{\max}$ time (though it may be disconnected at a given instant).

3. The introduction positions the work against methods that discard samples. However, it would be valuable to contrast this "memory of costs" approach with other memory-based ZO methods, such as those using momentum or filter-based techniques (like the cited Chen et al. [2022]). The proposed memory is on the raw function values $l(x_t + \epsilon d^j)$, not on the aggregated gradient estimate or the iterates themselves. Highlighting this distinction more clearly would further sharpen the paper's contribution.

A minor typo: (8) should have $z_t^j$.

---

> ### Author Rebuttal · Authors · 2025-07-29
>
> We are glad that you found our idea both intuitive and novel, and we truly appreciate your recognition of the rigor of our theoretical analysis. We sincerely thank you for your valuable suggestions and observations. In the following, we provide a detailed account of the changes we would make if the paper is accepted, as well as those we have already included in the revised version.
>
> We thank you for pointing out the interesting reference Karandikar et al. (2023) that we were not aware of and that we will include in our bibliography. As mentioned in your comment too, the method in the pointed reference needs $2D$ simultaneous function evaluations per iteration and, thus, it does not employ our memory paradigm to mitigate the number of these queries. Therefore, we really thank you for pointing out this reference as it shows that the idea of using a memory mechanism can be beneficial also in more general settings, as the stochastic approximation one considered in Karandikar et al. (2023). If the paper is accepted, we will explicitly discuss this aspect in the revised version of the paper.
>
> We thank you for raising this point about the potential tradeoff involving $T_{\text{max}}$. We clarify that $T_{\text{max}}$ denotes an upper bound on the number of iterations required to sample all directions at least once in the window $[t, t + T_{\text{max}} - 1]$. While a natural scenario may involve a cyclic and single sampling of the directions, $T_{\text{max}}$ is not necessarily equal to the number of directions $D$. Moreover, we emphasize that the formalization of our general estimation scheme (see Assumption 2) does not imply that increasing the number of directions $D$ improves gradient estimation accuracy at a given point. For example, when using the canonical basis, once all basis directions have been sampled, adding more directions does not yield a better estimate. To conclude, there are not tradeoffs since, in the absence of specific constraints, the most effective strategy is to cyclically sample the available directions, thereby minimizing $T_{\text{max}}$ subject to the scenario’s sampling budget (e.g., two points per iteration). If the paper is accepted, we will provide some insights about this aspect in the revised version of the paper.
>
> We thank you for the suggestions about comparisons with filter-based techniques. In the numerical simulations, we accordingly performed novel tests to show that our algorithm outperforms the method by Chen et al. (2022) in terms of convergence speed. We already included these tests in the revised version of the paper.
>
> We thank you carefully reading our paper and spotting the typo in Equation (8).
>
> We thank you for the question about a stochastic sampling extension. Extending Assumption 3 to a stochastic setting technically translates in considering stochastic timescale separation rather than deterministic timescale separation. More in detail, the technical assumption is that we would need a strictly positive expected value $\mathbb{E}[s_t^j]$ of each selector $s_t^j$. With this assumption at hand, we should be able to get a stochastic counterpart of the current main result with almost sure guarantees. In other words, we should rely on the concept of almost sure convergence rather than on the concept of asymptotic stability. If the paper is accepted, we will better clarify this point in the revised version of the paper.
>
> We sincerely thank you for your question regarding momentum-based methods and extensions to other scenarios without strong convexity. It helped us to realize that the modularity enabled by our timescale separation perspective naturally paves the way for extending our meta-algorithm to (i) more advanced variants, for example, by using accelerated methods as a baseline instead of plain gradient descent, and (ii) problems with more class of functions. Indeed, timescale separation introduces modularity in the sense that only the reduced system analysis (in the current version, the gradient descent analysis in strongly convex setups) needs to be adapted to the new baseline (for instance, an accelerated method) and setup (for instance, nonconvex functions), while the rest of the framework remains unchanged. If the paper is accept, we will explicitly mention this observation in the revised version. We agree with your observation: in the nonconvex case, we should lose linear convergence (unless one considers Polyak–Łojasiewicz functions). More specifically, convergence should be analyzed by using LaSalle’s Invariance Principle to establish convergence to the set of stationary points, rather than relying on Lyapunov-based theorems, which are suited for the stability of a single equilibrium point. If the paper is accepted, we will explicitly discuss these aspects in the revised version of the paper.
>
> We agree with your observation on the memory tradeoff. In one sentence, our method trades memory for convergence speed.
>
> Thank you again for your patience and effort in reviewing our paper!

---

> > ### Comment · Reviewer_p5uW · 2025-08-03
> >
> > Thank you for addressing most of my concerns.
> >
> > Regarding the work by Karandikar et al., to the best of my understanding, while they require $2D$ function evaluations, it is still linear in $D$ under a more generalized, stochastic setting. It is unclear at this point whether the proposed memory paradigm can be extended to more general case without requiring as many function evaluations.
> >
> > Extensions to PL functions should relatively be straight forward, and most of the Lyapunov based arguments will follow as is. Instead of using LaSalle's invariance, considering time-dilated coordinate transformation may possibly yield easier proof strategy.
> >
> > I thank the authors for running the additional experiments and for detailed clarification in general. Based on the clarifications and discussion with other reviewers, I will suitably update my rating.

---

> > > ### Author Response · Authors · 2025-08-03
> > >
> > > We thank you again for your comments and suggestions. Below, we provide a couple of further remarks.
> > >
> > > Regarding the work by Karandikar et al., we believe that our memory mechanism could also be beneficial in their setting. Their method uses a finite number of function evaluations per iteration, and thus, incorporating our memory mechanism could potentially replicate their approach with a reduced number of evaluations (e.g., only a single evaluation per iteration), while preserving the convergence properties established in their work.
> > >
> > > We also thank you for the suggestion about the time-dilated coordinate transformation; we will definitely investigate this approach as a possible avenue for extensions beyond the strongy convex and PL settings.

---

> > > > ### Comment · Reviewer_p5uW · 2025-08-04
> > > >
> > > > Thank you for the clarification. I am inclined to raise my score.

---

### Official Review · Reviewer_xBVs · 2025-06-30

**Clarity:** 2
**Significance:** 3
**Originality:** 3
**Rating:** 5
**Confidence:** 3

**Summary:**

The authors propose a zeroth-order gradient method for unconstrained convex optimization when the gradient of the objective (or cost) is unavailable, but can be estimated at a point $x$ using cost evaluations at perturbations of $x$ with respect to a finite and fixed set of directions. Precisely, they assume that the gradient of the cost, $\nabla l(x) = \sum_{j=1}^{D}g_{\epsilon}(l(x+\epsilon d^j),d^j) - e_{\epsilon}(x)$ for known directions $d^1,d^2,\cdots,d^D$. In the setting where it is not possible to simultaneously use all the perturbation directions in each optimization step, so that cost evaluations $l(x_t+\epsilon d^j)$ at a current solution $x_t$ is not available for $j=1,\cdots,D$, they propose a descent method that reuses cost evaluations from previous optimization steps to estimate the gradient $\nabla l(x_t)$ in round $t$. Then, under three main assumptions: strong-convexity, smoothness and an essentially-cyclic condition, they prove linear convergence of the last-iterate to a unique optimal solution. Their proof follows from an involved analysis of the descent method as a two-timescale dynamical system.

**Questions:**

### Some questions
1. Can the authors elaborate on what they mean by “model-free” in the context of their paper?
2. Just for clarity, can the authors also explain the essentially-cyclic condition in Assumption 3 in lay terms, and how this helps in the analysis?
3. The algorithm is easy to follow, but I had a conflicting opinion about the definition of the auxiliary variables in Equation 7. Consider the intended descent step $x_{t+1} = x_{t}-\gamma\nabla l(x_{t})$. Under Assumption 2, the previous expression would be equivalent to $x_{t+1} = x_{t}-\gamma\bigg(\sum_{j=1}^{D}g_{\epsilon}(l(x_t+\epsilon d^j),d^j) - e_{\epsilon}(x_t)\bigg)$. However, since the error is unknown and cost evaluations may not be available, the authors propose the update $x_{t+1} = x_{t}-\gamma\sum_{j=1}^{D}g_{\epsilon}(z_t^j,d^j)$ in Equation 8 where $\\{ z\_t^{j} \\} \_{j=1}^{D}$ are introduced as auxiliary variables with $z_t^j$ storing the cost evaluation along direction $d^j$ if available at step $t$. My confusion lies with the fact that $z_t^j$ is defined in Equation 7 to collect cost samples at $x_{t-1}$ i.e $z_t^j=l(x_{t-1}+\epsilon d^j)$. Meanwhile, I expected that $z_t^j=l(x_t+\epsilon d^j)$ so that in the best case: when all the cost samples are available and Assumption 2 holds exactly (with no errors), $g_{\epsilon}(z_t^j,d^j)=g_{\epsilon}(l(x_t+\epsilon d^j),d^j)$ and Equation 8 simply executes a true gradient descent update.

    It is interesting that the analysis still seems to check out nonetheless. Precisely, $z_t^j=l(x_t+\epsilon d^j)$ is the equilibrium point of the dynamic system formed by definition of the auxiliary variables in Equation 10b.

4. L260: Is this expression $F(0,t)=0$ obvious. Also what does $\xi=0$ mean? It would help if there is some distinction between scalar and vector variables in the paper, for example the real value $0$ and vector of all zeros.
5. L535: I assume the authors also use the expression in equation 14?

### Some typos
1. L109: is this missing an index? The LHS is a vector while the RHS is a scalar value.
2. L132: Do you mean $l(x_t,\epsilon d^j)$ instead of $\nabla l(x_t,\epsilon d^j)$.
3. Equation 8: Should be $z_t^j$ instead of $z_t$.
4. Equation 12b: Should be $\tilde{x}_t$ instead of $\tilde{x}$
5. Equation 24: $V$ is a multi-variable function by definition but is defined here with single input variables.
6. L255: $\xi^t$ should be $\xi’$.
7. Equations 35 and 36: These are exactly the same expression.
8. L540: you mean $L$-smooth?

**Ethical Concerns:**

["NO or VERY MINOR ethics concerns only"]

**Final Justification:**

I maintain my positive score of the paper due to the listed strengths and satisfactory discussion with the authors. In particular, during the discussion period the authors 1) proposed to address the listed weakness with experiments to empirically compare their method with relevant related works, and 2) discovered an alternative version of the algorithm with more intuitive descent steps at little-to-no cost to the analysis.

**Quality:**

3

**Strengths And Weaknesses:**

### Strength

The authors address an interesting problem in unconstrained convex optimization where the cost function and its gradient are unknown, however cost evaluations along a finite and fixed set of directions, when available, can be used to estimate the gradient. Moreso, their two timescale interpretation of the resulting descent method, and its analysis as a fast vs slow dynamical system, are quite interesting. Finally, from a quick glance at the technical content in the main text seem accurate.

### Weakness
The paper is motivated by claims that multi-point zeroth other methods are too expensive, while existing single-point methods suffer from slow convergence. However, there is no mention of what kind of rates have been achieved in prior works. To properly validate the authors claim of “faster convergence” with their method, I believe it will be helpful to reference notable convergence rate results in related literature especially from Chen et al. 2022, Xiao et al 2023 and Zhang et al 2024 – which reportedly already achieve improved performance in terms of convergence of single-point methods.

---

> ### Author Rebuttal · Authors · 2025-07-29
>
> We thank you for this exhaustive and positive feedback on our work, we are happy that you found the problem we address interesting and that you appreciated the two-timescale interpretation of our method. We really thank you for all the useful suggestions. In the following, we provide a detailed account of the changes we would make if the paper is accepted, as well as those we have already included in the revised version.
>
> We thank you for the suggestion about comparisons with state-of-the-art algorithms. In the revised version of the paper, we added numerical experiments that show that our method sensibly outperform those proposed by Chen et al. (2022) and Zhang et al. (2024) in terms of convergence speed. We omitted the comparison with Xiao et al. (2023), as their method allows for possibly discarding a new cost sample at each iteration, which may offer an advantage in reducing the number of queries rather than accelerating convergence and, thus, our method would have unfairly outperformed it in terms of convergence speed. Although the aforementioned works only establish $\mathcal{O}(1/\sqrt{t})$ convergence rates, we chose not to explicitly claim the superiority of our exponential rate, as we focus on strongly convex problems, in contrast to their merely convex settings.
>
> We thank you for the question about "model-free" meaning. We refer to model-free (or black-box) optimization methods as those that do not require analytical knowledge of the objective function. If the paper is accepted, we will clarify this point in the revised version.
>
> We thank you for the question about the essentially cyclic rule. Assumption 3 ensures the existence of an upper bound $T_{\text{max}} \in \mathbb{N}$ (independent of $t$) on the number of iterations required to select all directions $d^j$ at least once in the interval $[t,t+T_{\text{max}} - 1]$.  In the analysis, this essentially-cyclic rule allows for establishing the exponential stability properties of the memory mechanism. If the paper is accepted, we will better clarify these aspects in the revised version of the paper.
>
> We sincerely thank you for your insightful observation regarding the best-case scenario where all samples are available. Your suggestion led us to consider an alternative version of our meta-algorithm, where the solution estimates are updated according to $x_{t+1} = x_t - \gamma \sum_{j=1}^{D}(s^j_t g_\epsilon(\ell(x_t + \epsilon d^j),d^j) + (1-s_t^j) g_\epsilon(z_t^j,d^j))$, which employs the same cost samples as the original version, but benefits from the improvement you pointed out. Importantly, the analysis remains unchanged, as it still fits within the timescale separation framework developed for the original version. Indeed, the memory mechanism is unaffected, and the reduced system continues to correspond to the standard gradient method.
> We have tested this alternative formulation and observed a slight performance improvement compared to the original version. A third version of the algorithm could instead use samples of the form $\ell(x_{t+1} + \epsilon d^j)$ instead of $\ell(x_t + \epsilon d^j)$ in the update of the auxiliary variables $z_t^j$.  From a technical standpoint, also this change does not affect the convergence analysis. However, the other two formulations offer the advantage of enabling a parallel update of the solution estimate $x_t$ and the auxiliary variables $z_t^j$ rather than a sequential one, which is particularly beneficial in large-scale applications where the problem dimension $n$ and/or the number of perturbation directions $D$ is large. If the paper is accepted, we will modify the algorithm design with the update described above. Once again, thank you for this valuable suggestion!
>
> Thank you for the questions about $F(0_{n + D},t) = 0_{n + D}$ and $\xi = 0_{n +D}$. First of all, in the revised version of the paper, we already followed your suggestion and used the notation $0_n$ to denote the vector of all zeros of dimension $n$. Since $\xi$ is the stack of the error between (i) the solution estimate and the problem solution $\theta_\star$ and (ii) the auxiliary variables and their equilibrium values, $\xi = 0_{n + D}$ means that the solution estimate corresponds to the problem solution $\theta_\star$ and that all the auxiliary variables are at their equilibrium values. Hence, the origin must be an equilibrium of the "nominal" system described by Eq. (19), i.e., $F(0_{n + D},t) = 0_{n +D}$.  If the paper is accepted, we will better highlight those steps to help grasping this result.
>
> We thank you for the question about L535. We use the steps $\sum_{\tau=t+1}^{t+T_{\text{max}}-1}\|\Phi(\tau,t)\tilde{z}\|^2 = \sum_{\tau=t}^{t+T_{\text{max}}-1}\|\Phi(\tau,t)\tilde{z}\|^2 - \|\Phi(t,t)\tilde{z}\|^2 = U(\tilde{z},t) - \|\tilde{z}\|^2$, where the last equality uses the definition of $U$.
>
> We finally thank you for your careful reading and for spotting so many typos. We accordingly fixed all the spotted typos except for the first one. Indeed, in L109, we are considering the scalar case $n=1$ and, thus, also the left-hand-side is a scalar value.
>
> Thank you again for your patience and effort in reviewing our paper!

---

> ### Comment · Reviewer_xBVs · 2025-08-03
>
> I thank the authors for providing more clarity on the paper and strongly support incorporating their comments (particularly on questions 1, 2 and 4) to the final version as this will definitely improve readability of the paper. Please see other comments below:
>
> > We thank you for the suggestion about comparisons with state-of-the-art algorithms. In the revised version of the paper, we added numerical experiments that show that our method sensibly outperform those proposed by Chen et al. (2022) and Zhang et al. (2024) in terms of convergence speed...
>
> I appreciate the authors addition of new experiments (comparing their method with Chen et al. (2022) and Zhang et al. (2024)) to help validate their claims of faster convergence in the paper. From a theory standpoint, I also strongly suggest highlighting the stated convergence rates early on in the paper (of course with the caveat that these works only consider convexity). This, in addition to the discussion on Xiao et al. (2023), will in the worst case help readers fully grasp the contributions of the paper.
>
> **Response to Author's comment on question 3**: Sorry I am just noting that there is a typo in my original question. Particularly, I meant that in round $t$, it made sense to compute the gradient evaluations (when possible) at $x_t$ rather than $x_{t-1}$. So, use $z_{t}^{j} = z_{t-1}^{j} + s_{t-1}^{j}\left(\ell\left(x_{t}+\epsilon d^{j}\right) - z_{t-1}^{j}\right)$ instead of $z_{t}^{j} = z_{t-1}^{j} + s_{t-1}^{j}\left(\ell\left(x_{t-1}+\epsilon d^{j}\right) - z_{t-1}^{j}\right)$ in the expression after Equation (7). This would reduce to what I wrote earlier if the function evaluations are available at step $t$ and definitely result in your alternative update. Apologies again for the confusion. That said, yes I understand why this more intuitive version should lead to performance improvements. It is interesting that the previous update works, because in the best case for a strongly convex function the updates would be using larger descent steps than required in each round and as a result may be unstable and never converge to an optimal solution. On another note, I am not sure about replacing $\ell\left(x_{t}+\epsilon d^{j}\right)$ with $\ell\left(x_{t+1}+\epsilon d^{j}\right)$ because then the updates become implicit and are not exactly practical. Finally regarding the analysis for the alternative version, I understand that the resulting dynamical system in 10b would be alternating rather than simultaneous. This would change the analysis in some ways, for example the fixed point in L181 is $\overline{z} = L_{\epsilon}(x_{t+1})$ rather than $\overline{z} = L_{\epsilon}(x_{t})$, but I also find it plausible that the analysis still checks out. Just for clarity, can the authors highlight notable changes in the analysis and how they handle them? Also, does this affect the main result in Theorem 1?
>
> **Response to Author's comment on L535**: The expression is indeed true and I am only making a subtle note regarding the second term before the last equality.
>
> **Response to typo on L109**: Thanks for clarifying! I missed the $n=1$ part.

---

> > ### Author Response · Authors · 2025-08-03
> >
> > First of all, we would like to thank you again for your time and for the kind and insightful discussion regarding our work.
> >
> > Upon re-reading our previous reply, we realized that we did not clearly explain the two alternative versions to the one proposed in the first submission of the paper, and, as a result, there was some misunderstanding that led to your comment about the implicit update. To avoid any ambiguity, we rewrite the alternative proposals as follows:
> >
> > Version a)
> >
> > $x_{t+1} = x_{t} - \gamma \left(\sum_{j=1}^D s_t^j g_\epsilon(\ell(x_t + \epsilon d^j), d^j) + (1 - s_t^j) g_\epsilon(z_j, d^j)\right)$
> >
> > $z_{t+1}^j = z_t^j + s_t^j (\ell(x_t + \epsilon d^j, d^j) - z_t^j), \forall j \in {1,\dots,D}$
> >
> >
> > Version b)
> >
> > $x_{t+1} = x_{t} - \gamma \sum_{j=1}^D g_\epsilon(z_j, d^j)$
> >
> > $z_{t+1}^j = z_t^j + s_t^j (\ell(x_{t+1} + \epsilon d^j, d^j) - z_t^j), \forall j \in {1,\dots,D}$
> >
> > Version a) is the one that we prefer, as it avoids the problem you highlighted in your first comment, while concurrently allowing for (possibly) updating $x_t$ and $z_t$ in a parallel fashion rather than a sequential one. Version b), instead, needs a sequential update and this may be a little disadvantageous in large-scale scenarios.
> >
> > As for the analysis, let us focus on version a) only. Our analysis is based on the generic timescale separation result in Appendix A. Technically, it consists in analyzing the boundary-layer system (the $z$ update with an arbitrarily fixed $x$) and the reduced system (the $x$ update with $z$ always in the parameterized equilibrium). We note that the $z$ update of version a) is exactly the same as that in the first version of the paper and, thus, the boundary-layer system analysis is the same and, thus, we still have parametrized equilibria $\bar{z} = L(x_t)$. As for the reduced system, we note that it still coincides with the one analyzed in the first submission of the paper.
> > Hence, from a constructive point of view, the analysis does not change at all. However, with an ad-hoc analysis, it should be possible to get more refined convergence rate results. Indeed, by looking at the $x$-update of version a), we note that the "wrong" terms are those with $s_t^j = 0$, while in the previous version the selectors $s_t^j$ do not affect the $x$-update  Hence, it should be feasible to formalize corollaries in which, by imposing more structure in the sampling sequence (e.g., a minimum number of directions active in each iteration) results in more refined convergence rates.
> >
> >
> > Comment on L535: Thank you, we now understand what you are pointing out. We agree with you, it would be better to add a sentence such as: “Let us check (15b), i.e., the decrease condition of the Lyapunov function $U$ along the trajectories of the boundary-layer system (14). To this end, by definition (58), we note that...” before the chain of equalities that concludes the proof.

---

> > > ### Comment · Reviewer_xBVs · 2025-08-04
> > >
> > > Dear Authors,
> > >
> > > Thanks for the further clarifications! I am satisfied with the responses and will maintain my positive score.
> > >
> > > Best regards.

---

### Decision · Program_Chairs · 2025-09-17

**Decision:**

Accept (poster)

**Comment:**

The paper introduces a simple yet effective memory mechanism for model‑free (zeroth‑order) optimization that reuses past cost samples to form weighted descent directions, which allows to achieve faster convergence without increasing the number of function queries per iteration. Reviewers consistently liked the novelty and elegance of the auxiliary‐memory idea, the rigorous two‑timescale dynamical‑systems analysis yielding linear convergence for strongly convex objectives, and the modest implementation overhead (only an
m‑dimensional vector). It seems that the authors’ rebuttal addressed all major concerns: they added experiments against additional state‑of‑the‑art baselines, expanded the empirical study to ridge regression and a broader range of problem dimensions, clarified theoretical assumptions (model‑free definition, essentially‑cyclic sampling), and proposed alternative algorithmic variants that improve practicality without altering the convergence proof. While the current theory is limited to strongly convex, the noiseless setting and the experimental suite is a little small, the combination of solid theory and clear empirical gains merits acceptance of the paper.